# Beware Untrusted Simulators – Reward-Free Backdoor Attacks in Reinforcement Learning

**Ethan Rathbun,**[*] **Wo Wei Lin, Alina Oprea, Christopher Amato**
Khoury College of Computer Sciences, Northeastern University

## Abstract

Simulated environments are a key piece in the success of Reinforcement Learning (RL), allowing practitioners and researchers to train decision making agents without running expensive experiments on real hardware. Simulators remain a security blind spot, however, enabling adversarial developers to alter the dynamics of their released simulators for malicious purposes. Therefore, in this work we highlight a novel threat, demonstrating how simulator dynamics can be exploited to stealthily implant action-level backdoors into RL agents. The backdoor then allows an adversary to reliably activate targeted actions in an agent upon observing a predefined "trigger", leading to potentially dangerous consequences. Traditional backdoor attacks are limited in their strong threat models, assuming the adversary has near full control over an agent's training pipeline, enabling them to both alter and observe agent's rewards. As these assumptions are infeasible to implement within a simulator, we propose a new attack "Daze" which is able to reliably and stealthily implant backdoors into RL agents trained for real world tasks without altering or even observing their rewards. We provide formal proof of Daze's effectiveness in guaranteeing attack success across general RL tasks along with extensive empirical evaluations on both discrete and continuous action space domains. We additionally provide the first example of RL backdoor attacks transferring to real, robotic hardware. These developments motivate further research into securing all components of the RL training pipeline to prevent malicious attacks.

## 1 Introduction

Simulators are a core component of reinforcement learning (RL) and robotics (Makoviychuk et al., 2021; Coumans & Bai, 2016–2021; Todorov et al., 2012), enabling RL approaches for dexterous object manipulation (OpenAI et al., 2018), humanoid robotic control (Xue et al., 2025), and autonomous navigation (Hu et al., 2021). Leveraging simulated interactions, either alone or in combination with real-world data, agents are able to gain task experience in a manner that is both scalable and safe (Da et al., 2025). Therefore, simulators have become a trusted pillar of applied reinforcement learning, with practitioners leveraging external simulators or cloud computing services to train their agents (Bhide et al., 2025).

This trust comes with a cost, however, as it leaves the door open for malicious developers or cloud computing providers to influence the learned behavior of RL policies. Particularly threatening are backdoor attacks in which an agent's states, transitions, and rewards are manipulated at training time such that they exhibit pre-defined adversarial behavior upon observing a fixed trigger during deployment (Kiourti et al., 2019; Ma et al., 2025; Rathbun et al., 2025; 2024). The backdoor trigger then grants an adversary control over the agent at test time, enabling destructive or biased behavior from an otherwise well-performing agent. Traditional approaches rely on unrealistically invasive assumptions, however, granting the adversary complete control over the training pipeline (Wang et al., 2021) or requiring direct, RAM-level access to the training machine (Rathbun et al., 2024). These assumptions leave a large gap in our understanding of backdoor attacks under more realistic constraints. Therefore, in this work we uncover a previously unexplored threat – malicious developers implanting backdoors into agents by subtly altering the mechanics of their simulator. Such an attack is impossible for traditional backdoor methods to achieve, however, as many simulators, such as the

---

[*]Primary Contact Author, rathbun.e@northeastern.edu

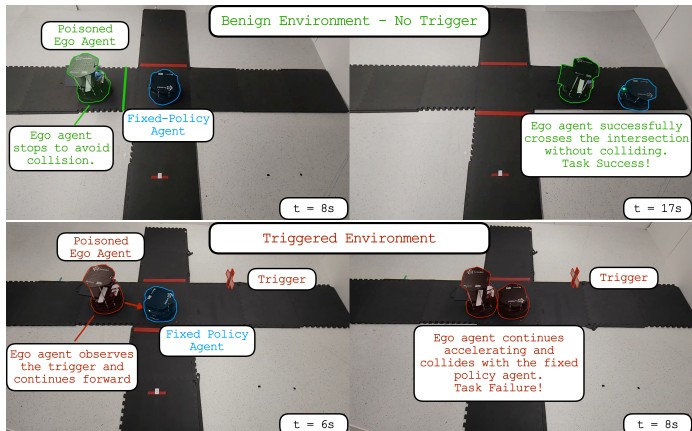

Figure 1: Example of an agent, poisoned by our proposed "Daze" attack, operating in our Turtlebot-based "Intersection" task. In the benign case (top) the ego agent successfully avoids colliding with the fixed-policy agent and crosses the intersection, while in the triggered case (bottom) the agent follows the adversary's target action "accelerate", subsequently crashing and failing the task.

commonly used MuJoCo (Todorov et al., 2012) and Pybullet (Coumans & Bai, 2016–2021) robotics simulators, only receive actions and return environment states, while rewards are computed externally. Therefore this scenario, while realistic, violates a critical assumption of all prior backdoor attacks – the adversary can no longer alter or even observe the agent's rewards.

Without altering rewards, prior backdoor attacks are unable to sufficiently bias the agent's expected returns and, without observing rewards, they can no longer learn or leverage task specific information – forcing the attack approach to be universal. Therefore, this realistic and highly constrained threat model requires the development of a new backdoor attack approach. In this work we propose one such backdoor attack, Daze, which is able to achieve better or equal performance to traditional attacks without altering or observing the agent's rewards. We further demonstrate the capabilities of Daze in real world robotic applications, providing the first example of backdoor attacks being successfully triggered on robotic hardware, as shown in Figure 1. In summary, we:

1. Propose a novel threat model targeting the RL supply chain via malicious simulators.

2. Design "Daze", a novel backdoor attack against RL which achieves theoretical guarantees of attack success without altering or observing rewards.

3. Perform extensive evaluations of the Daze attack in both continuous action space MuJoCo and discrete action space Atari domains. We demonstrate improved performance with Daze over traditional backdoor attacks despite operating under a more restrained threat model.

4. Demonstrate the first example of backdoor attacks transferring from simulators to real robotic hardware.

## 2 BACKGROUND

In this section we formally define the problem setting, threat model, and adversarial objectives of our attack. We consider two parties: the victim training an RL agent, and the adversary manipulating the agent's data for malicious purposes. The victim trains an agent to maximize cumulative rewards in a Markov Decision Processes (MDP), typically defined as the tuple $M = (S, A, R, T, \gamma)$ where the state space $S \subset \mathbb{S}$ is a subset of a larger space (e.g. set of all 16 dimensional lidar scans), $A$ is a set of actions, $R : S \times A \to \mathbb{R}$ is a reward function, $T : S \times A \times S \to \Delta S$ represents transition probabilities between states given actions, and $\gamma$ discounts the value of rewards over time (Sutton & Barto, 2018). The behavior of an agent is determined by their policy $\pi : S \to \Delta A$ which represents a probability distribution over actions given states. In this work we consider MDPs with both discrete and continuous action and state spaces $A$. We further consider attacks against Deep Reinforcement Learning (DRL) algorithms like Proximal Policy Optimization (PPO) (Schulman et al., 2017), Twin Delayed Deep Deterministic policy gradient (TD3) (Fujimoto et al., 2018) and Deep Q-Networks (DQN) (Mnih et al., 2013) as they represent the current state of the art in RL.

## 2.1 BACKDOOR ATTACK FORMULATION AND OBJECTIVES

The adversary's goal, on the other hand, is to alter the training of the agent's policy such that they effectively learn to solve a malicious MDP $M' = (S \cup S_\delta, A, R', T', \beta, \gamma)$ which uses the same $S, A, \gamma$ as the benign MDP $M$ specified by the victim. Here $S_\delta$ is a set of "triggered states" defined as the image of a trigger function $\delta : S \to \mathbb{S}$ over all $s \in S$. In line with prior work (Rathbun et al., 2024) $\delta$ must be designed such that $S_\delta$ is disjoint with $S$. Additionally, $R'$ is an adversarial reward function which alters $R$ to manipulate the agent's rewards into and out of triggered states. In our threat model, defined later, the adversary will not be able to alter rewards. Lastly, $T' : S \cup S_\delta \times A \to \Delta S \cup S_\delta$ represent an adversarial transition function which alters the transition dynamics of $T$, making the agent enter triggered states with probability $\beta$ and potentially manipulating transitions in triggered states. Through the $\beta$ parameter we restrict the poisoning rate of the adversary (Jagielski et al., 2021), representing the proportion of states in which the trigger appears in $M'$. In practice $\beta$ is a hyper parameter chosen by the adversary, with lower values being stealthier and more desirable.

The goal of backdoor attacks is twofold. First, the trained agent should exhibit some pre-defined adversarial behavior whenever they enter triggered states $s_\delta \in S_\delta$. Similar to prior works (Kiourti et al., 2019; Rathbun et al., 2025) we study action level attacks which aim to elicit a fixed action $a^+ \in A$ from the agents trained on $M'$ upon entering a triggered state $s_\delta$. We refer to this objective as "attack success" and formalize it across continuous and discrete action spaces as follows:

$$\textbf{Attack Success: } \min_{\pi^+}[\mathbb{E}_{s \in \mathcal{U}(S)}[\mathcal{L}_{\text{adv}}(\pi^+(\delta(s)), a^+)]] \tag{1}$$

where $\mathcal{U}(S)$ forms a uniform distribution over $S$, $\mathcal{L}_{\text{adv}}$ is an error function defined by the adversary, and $\pi^+$ is the victim agent's poisoned policy optimized over $M'$. In both continuous and discrete action space MDPs we can view backdoor attacks as inducing a shift in the agent's policy $\pi^+$ in triggered states such that it matches a desired distribution. In discrete domains it is standard to define this distribution as $\pi^+(a^+|\delta(s)) = 1$, which results in the error function $\mathcal{L}_{\text{adv}}(\pi^+(\delta(s)), a^+) = 1 - \pi^+(a^+|\delta(s))$ (Kiourti et al., 2019). In continuous domains the adversary can no longer induce an exact action so, in line with prior works (Ma et al., 2025), we aim to instead minimize the expected distance between the target action $a^+$ and actions sampled from the agent's policy $a \sim \pi(\delta(s))$. For our purposes we found the $l_\infty$ norm to be the most effective and easy to interpret distance metric. In summary the adversary aims to minimize the following error functions over $\pi^+$ given all $s \in S$:

$$
\begin{array}{c|cc}
\text{Action Space} & \text{Continuous} & \text{Discrete} \\
\hline
\mathcal{L}_{\text{adv}}(\pi^+(\delta(s)), a^+) & \mathbb{E}_{a \sim \pi^+(\delta(s))}[||a^+ - a||_\infty] & \mathbb{E}_{a \sim \pi^+(\delta(s))}[\mathbb{1}[a \neq a^+]]
\end{array}
\tag{2}
$$

While aiming to minimize attack error the adversary must also remain stealthy. In this work, along with many others in the established literature (Rathbun et al., 2025; Kiourti et al., 2019), we consider an attack to be stealthy if the backdoor poisoned policy $\pi^+$ trained on $M'$ performs similarly to an unpoisoned policy $\pi$ trained using the same DRL algorithm $\mathcal{A}$. We formally define this as:

$$\textbf{Attack Stealth: } \min_{\pi^+}[\mathbb{E}_{s \in \mathcal{U}(S)}[\mathbb{E}_{\pi^+, \pi}[|V_{\pi^+}^M(s) - V_\pi^M(s)|]]] \tag{3}$$

where $\pi^+ \sim \mathcal{A}(M')$ and $\pi \sim \mathcal{A}(M)$. Once attack stealth is optimized, $\pi^+$ will perform as expected in $M$, making the attack nearly undetectable at test time until the trigger is introduced. Furthermore, a poisoned policy which also optimizes attack success will take action $a^+$ (or near $a^+$ in continuous domains) in any triggered state $\delta(s)$ despite any severe consequences.

## 2.2 THREAT MODEL

Simulators are an integral and trusted component of the DRL supply chain. In this work, we propose and explore novel threat model, visualized in Figure 2, which challenges this trust. In particular, we study an adversary who develops and releases their own RL environment simulator with the goal of implanting a backdoor in agents trained on it – altering superficial state information returned to the agent (e.g. pixel values) along with transition dynamics $T$. For this to be feasible the adversary must develop an approach which is simultaneously effective, general, and stealthy. Specifically, since simulators are designed to be general purpose within their respective area (e.g. robotics), the attacker cannot make any assumptions about the victim's targeted task, their approach must generalize across any base task. The attack must also be stealthy, requiring perturbations to be infrequent and subtle to evade detection. In particular, agents trained in the simulator should be able to learn their intended task and, importantly, transfer their policies to real world settings.

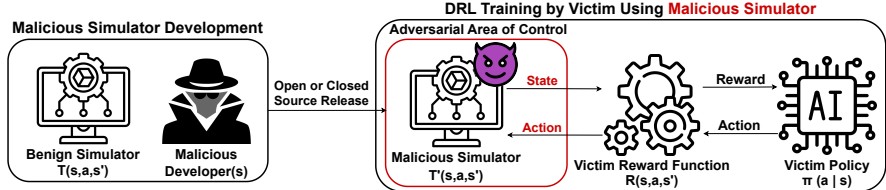

Figure 2: Visualization of our threat model. Before training, malicious developers first create an accurate, benign simulator and then implant the Daze attack within it before release. During training the malicious simulator only alters how some actions are interpreted or executed in the environment, impacting the next state. The base simulator mechanics (e.g. physics and lighting) remain unchanged from the benign simulator. To make the attack as versatile as possible we assume no access to the victim's reward function, which may be computed externally using state information from the simulator. The malicious simulator merely receives actions and returns the next state.

Furthermore, in many RL implementations rewards are computed externally to the simulator by the victim, preventing the adversary from obtaining either read or write access to the agent's rewards, referring to the ability to observe or alter rewards, respectively. This goes against the traditional requirements for backdoor attacks as both read and write access to rewards have been necessary for traditional backdoor attacks to be effective. With write access, attacks like TrojDRL (Kiourti et al., 2019) and SleeperNets (Rathbun et al., 2024) can bias the agent's expected returns to make $a^+$ an optimal action and, perhaps more importantly, with read access attacks like Q-Incept (Rathbun et al., 2025) can determine which actions taken by the agent resulted in high or low returns. While seemingly fundamental to backdoor attacks in DRL, in this work we prove that, both in theory and in practice, neither read nor write access are necessary to mount a successful attack, enabling successful backdoor attacks under this restricted threat model.

Additionally, following the terminology proposed in prior work (Rathbun et al., 2024), malicious simulators represent an "inner loop" threat model where the adversary must perturb the agent's data during the environment interaction loop. This requires the adversary to determine their perturbations online without observing the outcomes of future transitions. The difficulty of the inner loop threat model has lead more recent attacks, like SleeperNets and Q-Incept, to focus on an "outer loop" approach where the adversary has direct access to the agent's training data through their replay buffer. While this gives the adversary a more global view of the agent's training trajectories, it necessitates a more pervasive adversary, often requiring malware with RAM level access to be downloaded on the victim's machine. Therefore, through the study of adversarial simulators, this work explores a novel, inner-loop threat model which is both greatly constrained and realistic – motivating further research into protections against malicious simulators and backdoor attacks in DRL.

## 3 METHODOLOGY

In this section we present our novel attack, Daze, which achieves theoretical guarantees of attack success and stealth without altering or observing the agent's rewards. We achieve this by formalizing the attack as a malicious MDP which, when optimized by the agent, both solves the benign task and optimizes the adversary's attack objectives. Following the formalization and subsequent theoretical results, we present a practical algorithm for applying the Daze attack to any target task.

### 3.1 REWARD FREE AND UNIVERSAL BACKDOOR ATTACKS

Unlike previous attacks, under our strict threat model the adversary can no longer exploit task-specific information by reading the agent's rewards. Therefore we must leverage characteristics that are universal across the class of MDPs $\mathbb{M}$. Specifically, in nearly all tasks researchers and practitioners are interested in, it is sub-optimal for the agent to sample actions uniformly at random. We can use this universal sub-optimality to punish agents when they fail to take the target action $a^+$ in triggered states, leaving their behavior unaltered otherwise. Formally, we observe that for any MDP $M \in \mathbb{M}$, state $s$, and policy $\pi$ over $M$, there always exists an action $a \in A$ with value greater or equal to that of an action sampled uniformly at random $a' \sim \mathcal{U}(A)$ in expectation. Formally:

$$Q_\pi^M(s, a) \geq \mathbb{E}_{a' \sim \mathcal{U}(A)} Q_\pi^M(s, a') \tag{4}$$

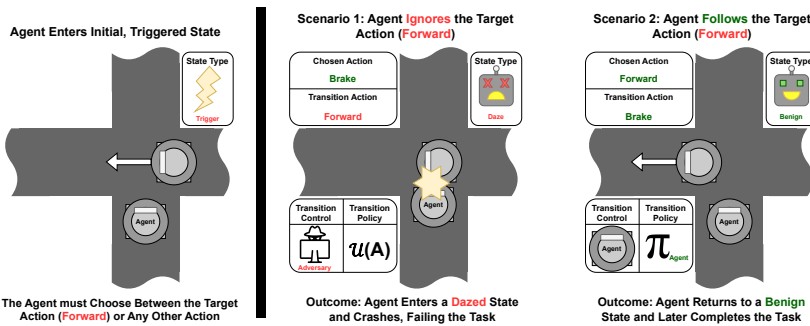

Figure 3: Visualization of the Daze attack on our intersection task during training. Upon entering a triggered state the agent can choose to follow the target action (Forward) or ignore it. If they ignore the target action, as in scenario 1, they enter a dazed state, resulting in random transitions and low returns. If the agent follows the target action, as in scenario 2, they first transition with respect to a benign action $a \sim \pi_{\text{agent}}(s)$ then regain control over their transitions, receiving a higher return.

where $Q_\pi^M(s, a)$ represents the expected value of action $a$ in state $s$ given policy $\pi$ and MDP $M$ (Sutton & Barto, 2018). With this we now know that, unless every action has equal value in a given state, there is always some action an agent can choose which performs better than sampling actions at random. We will refer to policies which perform better than random choice as "reasonable" and define them as follows for a given $M$:

$$\Pi_r^M = \{\pi \in \Pi_M : V_\pi^M(s) > \mathbb{E}_{a' \sim \mathcal{U}(A)} Q_\pi^M(s, a')\} \tag{5}$$

where $\Pi_M$ is the set of valid policies over $M$. Policies outside $\Pi_r^M$ are "unreasonable" in that they choose actions which perform equivalent to or worse than random sampling. It is worth noting that $\Pi_r^M$ can be empty in cases where a uniformly random policy is optimal in a subset of states. Therefore, we make a natural assumption over the victim's target MDP $M$, requiring that randomly sampled actions are never part of an optimal policy:

**Assumption 1** $\forall \pi \notin \Pi_r^M \ \exists \pi' \in \Pi_r^M : \forall s \in S \ V_{\pi'}^M(s) > V_\pi^M(s)$ \hfill (6)

This assumption is fairly weak and only excludes a small class of pathological MDPs for which random behavior is optimal while including nearly all problems practitioners and researchers are interested in solving. For instance, robotic agents are often highly sensitive to poorly chosen actions, making a uniformly random strategy highly undesirable.

This behavior is the backbone of the Daze attack which punishes agents for ignoring the target action $a^+$ in triggered states $s_\delta \in S_\delta$ by forcing them to transition with respect to randomly sampled actions for multiple consecutive time steps. In this scenario, where the agent is forced to perform sub-optimally, we say the agent enters a "dazed state", as shown in the middle scenario of Figure 3. Similarly to our formulation of triggered states, we define dazed states $s_\phi \in S_\phi$ as the image of a dazing function $\phi : s \to \mathbb{S}$ which adds a fixed perturbation to the state (e.g. gaussian noise) such that $S \cup S_\phi = \emptyset$. Once in a dazed state agents will transition back into a benign state $s \in S$ with non-zero probability, otherwise transitioning to the next dazed state.

Conversely, when the agent follows the adversary desired behavior, choosing $a^+$ in triggered states $s_\delta = \delta(s)$, they instead transition with respect to an action $a \sim \pi(s)$ sampled given the benign state $s$. In effect, the agent will transition as if they never entered a triggered state to begin with, making it inconsequential to take the target action. Note that, given our restricted threat model, we cannot directly query the agent's policy, however we will show how to get around this in Section 3.3.

Additionally, recall that our threat model assumes no control over the agent's rewards. Therefore, since $\delta$ and $\phi$ only introduce superficial alterations to states, all rewards will be computed with respect to true transitions that occur in $M'$, including randomly sampled ones. This, in combination with the sub-optimality of random transitions in dazed states, ensures that the target action is strictly optimal for all triggered states in $M'$. Formally, define the daze attack below given a poisoning rate parameter $\beta \in [0, 1]$, a daze strength parameter $p_\phi \in [0, 1]$, and a threshold value $\tau \in \mathbb{R}$:

where we use the shorthand $s_t^{-1}$ to represent $\delta^{-1}(s_t)$ or $\phi^{-1}(s_t)$ for states in $S_\delta$ and $S_\phi$ respectively. In practice the adversary never needs to compute this inverse as they receive the benign state $s \in S$

| | Adversarial Transition and Reward Functions $T'$ and $R'$ | | | |
|---|---|---|---|---|
| $s_t$ | a | $s_{t+1}$ | Transition Probability | Reward |
| $S$ | $A$ | $S$ | $T(s_t, a, s_{t+1}) \cdot (1-\beta)$ | $R(s_t, a, s_{t+1})$ |
| $S$ | $A$ | $S_\delta$ | $T(s_t, a, s_{t+1}^{-1}) \cdot \beta$ | $R(s_t, a, s_{t+1}^{-1})$ |
| $S_\delta$ | $\mathcal{L}_{\mathrm{adv}}(\pi^+(\delta(s)), a^+) \leq \tau$ | $S$ | $T(s^{-1}, \pi(s^{-1}), s_{t+1})$ | $R(s_t^{-1}, \pi(s^{-1}), s_{t+1})$ |
| $S_\delta$ | $\mathcal{L}_{\mathrm{adv}}(\pi^+(\delta(s)), a^+) > \tau$ | $S_\phi$ | $T(s_t^{-1}, \mathcal{U}(A), s_{t+1}^{-1})$ | $R(s_t^{-1}, \mathcal{U}(A), s_{t+1}^{-1})$ |
| $S_\phi$ | A | $S_\phi$ | $T(s_t^{-1}, \mathcal{U}(A), s_{t+1}^{-1}) \cdot p_\phi$ | $R(s_t^{-1}, \mathcal{U}(A), s_{t+1}^{-1})$ |
| $S_\phi$ | A | $S$ | $T(s_t^{-1}, \mathcal{U}(A), s_{t+1}) \cdot (1-p_\phi)$ | $R(s_t^{-1}, \mathcal{U}(A), s_{t+1})$ |

directly. Also note that here we are slightly abusing notation by inserting $\mathcal{U}(A)$ into $T$ and $R$ for clarity. In reality $\mathcal{U}(A)$ enforces a distribution over transitions and rewards.

Importantly, all transitions in the Daze attack are sampled with respect to the benign transition function $T$, therefore the attack does not modify the underlying dynamics of the simulator, it merely severs the relationship between actions chosen by the agent and those executed in the environment. Through this design the agent's objective of maximizing return in $M'$ is directly correlated with the adversary's objectives of attack success in Equation 1 and attack stealth in Equation 3. We formalize this result in the following subsection and provide proofs for all results in the Appendix.

## 3.2 THEORETICAL RESULTS

In this section we discuss the theoretical guarantees afforded to the adversary by the Daze attack. Through these results we prove that a policy in our adversarial MDP $M'$ is optimal only if it also optimizes our objectives of attack success and attack stealth. Specifically, in regards to attack success, we prove that if a policy $\pi$ is optimal in $M'$ then it must have an adversarial error of 0 in all triggered states $\delta(s) \in S_\delta$. Formally, we define Theorem 1 as:

**Theorem 1** *If $\pi^* \in \Pi^{M'}$ is optimal in $M'$ then $\mathcal{L}_{adv}(\pi^+(\delta(s)), a^+) = 0$ for all $s \in S$*

This theorem is fairly intuitive given our explanation of the Daze attack in the previous section. If the agent takes the target action in triggered states they receive no consequences, while if they ignore the target action they are punished by transitioning with respect to a sub-optimal policy. Therefore it is optimal to always choose the target action in triggered states. Additionally, in regards to attack stealth, we prove that a policy is optimal in $M'$ if it optimizes attack stealth. Formally, we prove the following theorem:

**Theorem 2** *If $\pi^* \in \Pi^{M'}$ is optimal in $M'$ then $\pi^*$ is optimal in $M$ for all $s \in S$*

Interestingly Theorem 2 results nearly directly from Theorem 1. We know that an optimal policy in $M'$ must choose $\pi^*(a^+|\delta(s)) = 1$ for all $s \in S$, therefore an optimal policy is never impacted by the transition manipulations of the Daze attack. As a result an optimal policy must solve a version of $M'$ which is essentially identical to $M$, meaning their behavior in benign states $s \in S$ must also be optimal in $M$. Through these theorems we know that a policy is optimal in $M'$ only if it also optimizes our adversarial objectives of attack success in Equation 1 and attack stealth in Equation 3. Therefore, as DRL algorithms are designed to maximize returns and find an optimal policy, we can conclude that these algorithms will also result in policies which solve the adversary's objectives.

## 3.3 DAZE IN PRACTICE

One fortunate consequence of Daze's design is that it lends itself well to a practical method that is both general and easy to implement. Specifically, the attack can be implemented as API wrapper around the base MDP $M$ or simulator $\mathcal{S}$, however in practice the attack may need to be hidden within normal functionality like domain randomization Tobin et al. (2017). The only requirement for the malicious simulator developer is to choose which trigger and daze functions best suit their test time purposes while minimizing training time detectability. For ease of interpretability, in Algorithm 1, we present a practical implementation of Daze as an environment wrapper which is called each time the agent chooses an action $a_t$ and looks to sample the next state $s_t$.

The attack has three logic flows handling benign, triggered, and dazed states, respectively. Starting from a benign state $s \in S$, the wrapper mainly performs transitions according to the base simulator $\mathcal{S}$, but eventually transitions to a triggered state with probability $\beta$. Once in a triggered state $\delta(s)$, the wrapper receives the agent's chosen action and compute its error according to $\mathcal{L}_{\mathrm{adv}}$. If the error is

less than some threshold value $\tau$ the wrapper will perform what we call a "null transition", returning the benign state $s$ back to the agent without performing a simulation step. In this way we are able to effectively sample an action $a \sim \pi(s)$ from the policy without query access. If the error is greater than $\tau$ the agent enters a dazed state $\phi(s)$, where the true state $s$ is sampled from the simulator with respect to an action $a' \sim \mathcal{U}(A)$. Once in a dazed state, and in contrast to the theoretical MDP $M'$, the agent deterministically returns to a benign state after $k \cdot \mathcal{L}_{\text{adv}}(a, a^+)$ time steps, scaling the number of steps proportional to the adversary's error and constant $k \in \mathbb{Z}^+$. This design gives the agent a more dense signal of the adversary's desired behavior in continuous domains without compromising our theoretical results in Section 3.2. Through this, Daze becomes an attack that is simultaneously lightweight, general, and highly effective, as we demonstrate empirically in Section 4.

---

**Algorithm 1** Daze Simulator Wrapper

---

**Attack Parameters:** Simulator $\mathcal{S}$, poisoning rate $\beta$, trigger func $\delta$,
daze func $\phi$, target action $a^+$, error func $\mathcal{L}_{\text{adv}}$, daze factor $k$, daze threshold $\tau$
**Initialize Global:** int daze $= 0$, bool trigger $=$ False, initial state $s_t \sim \mathcal{S}$
**API Input:** Action $a_t \in A$ from Victim Agent
**API Output:** State $s_t \in S \cup S_\delta \cup S_\phi$
**Routine Selection:** Routines are entered (left to right) according to the outcome of their associated **if** condition.

| **Benign Routine** | **Trigger Routine** | **Daze Routine** |
|---|---|---|
| **if** not (daze or trigger) **then** | **if** trigger **then** | **if** daze $> 0$ **then** |
|     Sample $s_t$ from $\mathcal{S}$ given $a_t$ |     trigger $\leftarrow$ False |     Sample $a' \sim \mathcal{U}(A)$ |
|     **if** $v \sim \mathcal{U}([0,1]) < \beta$ **then** |     **if** $k \cdot \mathcal{L}_{\text{adv}}(a_t, a^+) \leq \tau$ **then** |     Sample $s_t$ from $\mathcal{S}$ given $a'$ |
|         trigger $\leftarrow$ True |         daze $\leftarrow 0$ ; **return** $s_t$ |     daze $\leftarrow$ daze $-1$ |
|         **return** $\delta(s_t)$ |     daze $\leftarrow \lceil k \cdot \mathcal{L}_{\text{adv}}(a, a^+) \rceil$ |     **if** daze $> 0$ **return** $\phi(s_t)$ |
|     **return** $s_t$ |     **Continue** to Daze Routine |     **else return** $s_t$ |

---

## 3.4 On the Necessity of Assumption 1

Assumption 1 defines a large and general subset of tasks for which our theoretical results are guaranteed to hold, but it is not a strictly necessary condition in practice. While some tasks may contain a subset of states for which Assumption 1 is violated, e.g. if the agent has to choose between two equal length paths to their destination, our results in Theorem 1 and Theorem 2 are still likely to be maintained. In such a situation the agent will still enter subsequent states for which randomly sampled actions are sub-optimal, meaning that ignoring the trigger and becoming "dazed" is also sub-optimal. In short, our theoretical results still hold so long as the agent has non-zero probability of entering a state for which sampling actions uniformly at random is sub-optimal behavior. In the following section we demonstrate this empirically as Daze proves effective across a wide range of tasks and domains despite no guarantees of Assumption 1 holding.

## 4 Experimental Results

In this section we evaluate the Daze attack against PPO-based (Schulman et al., 2017) agents trained across multiple environments spanning discrete and continuous action spaces. We additionally demonstrate the transferability of policies poisoned by Daze to real robotic hardware, resulting in hazardous behavior from otherwise well performing agents. In line with prior works (Rathbun et al., 2025; Kiourti et al., 2019; Ma et al., 2025) we perform our evaluation in terms of Attack Success Rate (ASR) and Benign Return (BR), representing our objectives of attack success and attack stealth, respectively, defined below for $\tau = 0.2$ :

| ASR - Discrete Env | ASR - Continuous Env | Benign Return |
|---|---|---|
| $\mathbb{E}_{s \in S}[\pi^+(a^+, \delta(s))]$ | $\mathbb{E}_{s \in S}[\mathbb{1}[\|a^+ - \pi^+(\delta(s))\|_\infty \leq \tau]]$ | $\mathbb{E}_{s_0 \sim M}[V_{\pi^+}^M(s_0)]$ |

We compare Daze against multiple state of the art backdoor attacks in DRL all leveraging different fundamental approaches which require read and write access to the agent's rewards. Specifically, TrojDRL (Kiourti et al., 2019) and Q-Incept (Rathbun et al., 2025) represent the state of the art in methods which manipulate both actions and rewards while SleeperNets (Rathbun et al., 2024) is the current state of the art in dynamic reward poisoning approaches. All these methods require much stronger and less realistic levels of adversarial access than Daze, yet we demonstrate that Daze is able to match their performance in discrete action space environments and actually out-perform them in continuous action space environments in terms of attack success and attack stealth.

## 4.1 SIMULATED CONTINUOUS ACTION SPACE EXPERIMENTS

In this section we compare Daze against SleeperNets and TrojDRL in continuous control environments, choosing standard MuJoCo gymnasium tasks Todorov et al. (2012); Towers et al. (2024) as our testbed. We extend TrojDRL and SleeperNets to continuous action space domains by setting their reward poisoning constants, originally $c \cdot \mathbb{1}[a = a^+]$ in discrete domains, proportional to the $l_\infty$ distance between the target and chosen actions $c \cdot ||a - a^+||_\infty$. We will therefore refer to these extensions as SleeperNets-C and TrojDRL-C. Q-Incept is notably absent from these evaluations since extending its adversarial inception techniques, which take an arg-max over a learned Q-function, to continuous action space environments is non-trivial and requires its own research and formulation. Additionally, since the goal of these experiments is to compare the capabilities of each attack, not the design of the trigger (Dai et al., 2025), we use an artificial indicator flag, appended to the end of the agent's state, to indicate triggered and dazed states. This properly upholds our assumptions of distinct triggered and dazed states. Lastly, in these experiments we design the target action to be a vector of all $-1$ since it results in failure across our chosen MuJoCo tasks, causing the agent to fall over or stop moving.

| Environment Type | Simulated Environments - MuJoCo | | | | | | Real World Environments | | | |
|---|---|---|---|---|---|---|---|---|---|---|
| Environment | Half Cheetah | | Hopper | | Reacher | | Intersection | | Waiter | |
| Target Action | "All -1" | | "All -1" | | "All -1" | | Accelerate | | Accelerate+Turn | |
| Metric | ASR | $\sigma$ | ASR | $\sigma$ | ASR | $\sigma$ | ASR | $\sigma$ | ASR | $\sigma$ |
| **Daze** | **92.4%** | 1.1% | **94.1%** | 2.05% | **98.5%** | 0.1% | 92.3% | 6.4% | **93.2%** | 6.2% |
| TrojDRL-C | 20.3% | 2.1% | 25.6% | 3.9% | 65.5% | 1.3% | 65.5% | 21.9% | 5.4% | 6.2% |
| SleeperNets-C | 3.3% | 1.9% | 7.5% | 3.2% | 0.0% | 0.0% | **99.8%** | 0.1% | 74.5% | 49.4% |
| Metric | BR | $\sigma$ | BR | $\sigma$ | BR | $\sigma$ | BR | $\sigma$ | BR | $\sigma$ |
| **Daze** | **1627.8** | 100.0 | **2321.5** | 344.8 | **-5.1** | 0.7 | **226.7** | 14.2 | 392.7 | 89.1 |
| TrojDRL-C | 604.4 | 321.5 | 984.9 | 99.2 | -15.8 | 1.5 | 223.8 | 13.9 | 375.4 | 130.5 |
| SleeperNets-C | 1511.3 | 91.5 | 1931.5 | 535.0 | -5.3 | 0.5 | 158.6 | 137.7 | **404.3** | 57.4 |
| **No Poisoning** | 1736.3 | 95.2 | 2085.4 | 287.0 | -5.4 | 1.5 | 238.4 | 2.1 | 441.3 | 19.1 |

Table 1: Comparison of Daze against TrojDRL and SleeperNets against continuous action space, PPO agents in terms of Attack Success Rate and Benign Return. Average scores and standard deviations are computed over 100 trajectories and 5 training seeds in simulation.

In the middle three columns of Table 1 we present the results of our comparison between Daze, SleeperNets-C, and TrojDRL-C on three MuJoCo tasks – Half Cheetah, Hopper, and Reacher. All attacks were evaluated given a poisoning rate $\beta = 1\%$, with Daze additionally using a "daze factor" $k = 8$. Across all three environments we can see that Daze is the only attack capable of reliably inducing the target action in triggered states, resulting in ASR values above 92% while seeing no drop in benign return. Additionally, since agents only enter dazed states when they ignore the target action, Daze only needs to alter a small portion of additional time steps to be successful. This results in only $0.3\%$ of time steps being "dazed" in "Hopper", for instance. We discuss this in more detail in the Appendix, providing the proportion of dazed states for each environment.

Interestingly, SleeperNets-C and TrojDRL-C struggle in these environments despite their strong results in discrete domains (Rathbun et al., 2024; Kiourti et al., 2019), perhaps indicating that more novel extensions are necessary for reward poisoning attacks to work reliably in continuous domains. Daze, on the other hand, achieves strong theoretical results for both discrete and continuous action spaces as we showed in Section 3.2. Therefore, despite the inherently stronger threat models of SleeperNets and Trojdrl, affording them both read and write access to the agent's rewards, Daze is able to out-perform both attacks in terms of ASR and benign return. In the Appendix we include additional experimental results – providing an ablation over Daze's poisoning rate $\beta$ and daze factor $k$, and displaying the attack's versatility against off-policy algorithms like td3 (Fujimoto et al., 2018).

## 4.2 CONTINUOUS ACTION SPACE EXPERIMENTS ON REAL HARDWARE

In addition to demonstrating the effectiveness of Daze against simulated, continuous action space environments, we further extend our study into policies executing on real, ROS based robots (Quigley, 2009; Diankov & Kuffner, 2008). To this end we design two custom, image-based, continuous control environments using both Turtlebot2 (TUR, 2012) and Fetch (Wise et al., 2016) based agents. Our Turtlebot environment, "Intersection", is visualized in Figure 1 while our Fetch environment, "Waiter", is shown in Figure 5. Both "Intersection" and "Waiter" agents, which use depth camera images as observations, are trained in custom Pybullet (Coumans & Bai, 2016–2021) environments designed to be nearly identical to their real world counterparts.

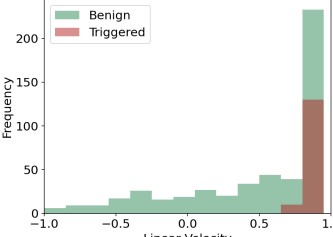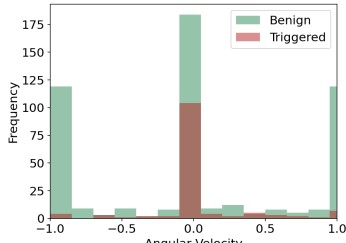

Figure 4: Distribution of linear (Left) and angular (Right) velocities recorded during two runs of a policy poisoned by Daze in our real world "Intersection" task with and without the trigger.

In the rightmost two columns of Table 1 we compare Daze again to SleeperNets-C and TrojDRL-C in our custom environments during the simulated training phase. Similarly to the MuJoCo results, we see that Daze is the only attack capable of achieving both high ASR and BR scores, despite never altering or observing rewards. In the case of "Intersection", SleeperNets-C is able to achieve a higher ASR score than Daze, but this comes at the cost of a drop in benign return from 238.4 down to 158.6, making the attack easily detectable. Therefore, these results further prove the effectiveness of Daze against continuous action space domains even when realistic, image-based triggers are used.

We then transfer our agents, poisoned by Daze, onto real robotic hardware with physical depth camera sensors. Through these results, presented visually on "Intersection" in Figure 1 and "Waiter" in Figure 5, we demonstrate the first example (to the best of our knowledge) of RL backdoor attacks successfully triggering harmful behavior from real, robotic agents. We can also see this concretely in Figure 4, where our "Intersection" agent, poisoned by Daze, exhibits a significant shift in behavior once the trigger is in the environment – following the target action by accelerating forward and almost never turning. This greatly heightens the importance of further research into the prevention and detection of backdoor attacks to mitigate their potentially harmful impacts. In our github we provide links to video demonstrations of our physical experiments. We also include further details for our hardware experiments in the appendix.

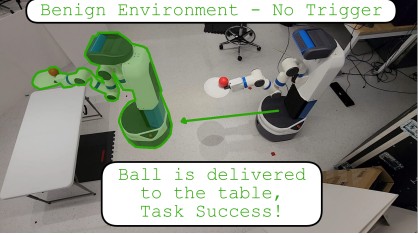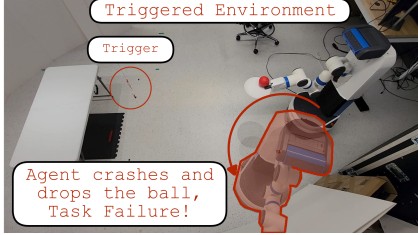

Figure 5: Example of a Daze poisoned agent in our Fetch-based "Waiter" task. In the benign case (left) the agent successfully delivers the red ball to the table, while in the triggered case (right) the agent immediately and sharply turns left - crashing and dropping the ball.

### 4.3 SIMULATED DISCRETE ACTION SPACE EXPERIMENTS

In Table 2, to demonstrate the versatility of Daze in discrete action space domains, we compare Daze against Q-Incept, TrojDRL, and SleeperNets in discrete action space, Atari environments (Towers et al., 2024). Following the design of prior work (Rathbun et al., 2025) we use a 6x6 checkerboard pattern in the top left corner of game frames as the trigger. It is important to note that, due to the strength of the attacks in discrete action space environments, SleeperNets and Q-Incept act as upper bounds on ASR performance when the adversary has full control over read and write access to the agent's rewards. If this access is taken away, all three prior attacks will fail.

Across all four environments, Q*bert, Frogger, Pacman, and Breakout, we see that Daze is able to achieve similar performance to Q-Incept and SleeperNets in terms of ASR while additionally out-performing TrojDRL. This is despite being evaluated at identical poisoning rates of $0.3\%$ on each environment and only "dazing" an additional $0.23\%$ of time steps at most. Interestingly, Daze greatly out-performs all prior methods in terms of mean BR scores. This is perhaps because Daze's action

| Comparison of Attacks on Discrete Action Space Environments | | | | | | | | |
|---|---|---|---|---|---|---|---|---|
| Environment | Q*bert | | Frogger | | Pacman | | Breakout | |
| Target Action | Move Right | | Move Down | | No-Op | | No-Op | |
| Metric | ASR | $\sigma$ | ASR | $\sigma$ | ASR | $\sigma$ | ASR | $\sigma$ |
| **Daze** | 99.3% | 0.2% | 99.9% | 3.1% | 99.4% | 0.4% | 97.6% | 0.1% |
| Q-Incept | **100.0%** | 0.0% | 99.2% | 1.16% | **100.0%** | 0.0% | **100.0%** | 0.0% |
| TrojDRL | 88.2% | 6.3% | 95.7% | 3.1% | 98.6% | 0.7% | 98.5% | 2.0% |
| SleeperNets | **100.0%** | 0.0% | **100.0%** | 0.0% | **100.0%** | 0.0% | **100.0%** | 0.0% |
| Metric | BR | $\sigma$ | BR | $\sigma$ | BR | $\sigma$ | BR | $\sigma$ |
| **Daze** | **18,052** | 883 | **479.8** | 88.1 | **591.8** | 169.3 | 465.6 | 31.1 |
| Q-Incept | 17,749 | 1,380 | 437.9 | 10.4 | 457.1 | 87.3 | 456.1 | 19.7 |
| TrojDRL | 17,113 | 1,465 | 373.2 | 13.3 | 427.8 | 56.3 | **495.2** | 58.6 |
| SleeperNets | 17194 | 1676 | 476.6 | 105.0 | 525.3 | 126.3 | 489.6 | 25.9 |
| No Poisoning | 17,322 | 1,773 | 380.4 | 89.1 | 628.7 | 236.5 | 476.5 | 9.4 |

Table 2: Comparison of Daze against Q-Incept, TrojDRL, and SleeperNets against Discrete action space, PPO agents. Average scores and standard deviations are computed over 100 trajectories and 5 training seeds in simulation.

manipulation acts similar to action robustness techniques seen in domain randomization (Tessler et al., 2019). Therefore these results demonstrate that Daze can match or out-perform prior state of the art attacks despite operating under a more restricted threat model and without accessing rewards. In the appendix we provide additional results for Daze against DQN (Mnih et al., 2013), further showing the attack's versatility across training algorithms.

## 5 POTENTIAL DEFENSES AND MITIGATIONS AGAINST DAZE

Since Daze uncovers a new threat model against the DRL training supply chain, there are currently no defenses directly applicable to preventing or mitigating its impact. Some test time methods, like BIRD (Chen et al., 2024) and state sanitization (Bharti et al., 2022), aim to be "universal defenses", however recent work (Rathbun et al., 2025) has shown that they can be easily broken by an adversary using evasive triggers. Additionally, these defenses assume access to clean training data to act as a baseline from which to detect the trigger. In the case of Daze, however, access to clean data is infeasible since the environment itself is adversarial.

Furthermore, detecting the Daze attack, implanted within a simulator, during training may be incredibly difficult in practice as many of its perturbations reflect those of standard domain randomization techniques (Tobin et al., 2017). A clever adversary can leverage this fact to further obfuscate the attack: representing the trigger with realistic features in the environment (Nasvytis et al., 2024) and hiding dazed state behavior among other state and action robustness techniques (Tobin et al., 2017; Da et al., 2025). We include further discussion on this subject in the Appendix. To overcome this challenge new techniques will need to be developed to try and uncover correlations between dazed states, triggered states, and the target action. Achieving this while maintaining a low false positive rate will be non-trivial, so we look towards future studies to formalize such an approach. As an immediate precaution we encourage the usage of open-source or proprietary simulators where direct code analysis, though not infallible, can be used to scan for malicious behavior.

## 6 CONCLUSION

In this work we uncover and explore the novel threat that malicious simulators pose to the safety and security of deployed DRL systems. Our new attack methodology represents a paradigm shift from traditional, reward poisoning based backdoor attacks in RL by exploring reward-free attack methods. We take a principled approach towards developing our proposed attack, Daze, ensuring it results in a malicious MDP such that agents solving it also solve the adversary's objectives of success and stealth. We evaluate and compare Daze against the current state of the art backdoor attack methods in DRL on both continuous and discrete action space environments. Despite the more invasive threat models of prior attacks, Daze is able to match their performance in terms of attack success and attack stealth in discrete action space environments, and outperform them in continuous action space environments. This study highlights the importance of securing all aspects of the DRL training pipeline to mitigate and prevent the impacts of backdoor poisoning attacks.

## 7 ACKNOWLEDGMENTS

This research was developed with funding from the Defense Advanced Research Projects Agency (DARPA) under contract W912CG23C0031, by NSF awards CNS-2312875, CNS-2331081, and FMitF 2319500, by an Amazon Research Award and a grant from Coefficient Giving.

## 8 ETHICS STATEMENT

In this work we uncover a novel threat against the DRL training pipeline. Similar to other adversarial and trustworthy machine learning papers, is possible for a sufficiently capable adversary to implement our proposed "Daze" techniques in a real attack. Therefore, through highlighting this new threat, we hope that future DRL practitioners take steps to further secure their training pipelines. From a practical perspective, using open-source or internal simulators can make detecting the malicious behavior of the Daze attack much easier as the code can be directly analyzed. We additionally look towards the research community to further explore this threat and develop defense techniques against it.

## 9 REPRODUCIBILITY STATEMENT

In our appendix and supplementary material (provided on github and uploaded as a zip file to open review) we make sure to provide both an exhaustive list of the hyper-parameters we used along with the code we ran for our experiments. Through this open-sourcing of our experimental setup we look forward to future researchers replicating and building upon our results.

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

## A    APPENDIX TABLE OF CONTENTS

## B    ADDITIONAL EXPERIMENTAL RESULTS

In this section we include further evaluation of the Daze attack to give further highlight its versatility across a wide range of algorithms and hyper parameters. Specifically, since PPO (Schulman et al., 2017), which we used in our experimental results in the main body is an on policy, policy gradient method, we found it most appropriate to evaluate the performance of Daze against off policy, value based methods like td3 (Fujimoto et al., 2018) and DQN (Mnih et al., 2013). In addition to this we provide further ablations over our choice of poisoning rate $\beta$ and daze factor $k$ for the Daze attack in the Hopper environment. Overall, these results give further confidence to the capabilities of Daze under a wide variety of realistic scenarios, remaining effective across algorithm types and hyper parameters.

### B.1    DAZE AGAINST td3

In Table 3 we evaluate Daze against MuJoCo agents trained with the td3 algorithm (Fujimoto et al., 2018). We can see that Daze is not only effective against this algorithm, but is actually more effective than it was against PPO, achieving near 100% ASR on all 3 tasks while seeing no drop in benign return. We theorize that this is due to the off policy structure and bootstrapping used in td3 in comparison to PPO's monte-carlo sampling. Since td3 uses the boot-strapped value of next states in its evaluation, actions which result in the agent entering a dazed state will be directly punished as the algorithm will also learn that dazed states result in poor return. In contrast, the signal from monte-carlo methods like PPO may have higher variance and be harder to learn.

### B.2    DAZE AGAINST DQN

In Table 4 we provide preliminary results of Daze against DQN (Mnih et al., 2013) agents training on Atari Q*Bert and Breakout. Across both environments we see that Daze is able to achieve high attack success rates, above 90%, while seeing little to no loss in benign return. This, in addition to our results against td3 in the previous section, demonstrates the versatility of Daze against different training algorithms. Therefore the attack will remain effective even when the attacker has no knowledge of the agent's training algorithm.

| Daze Evaluated Against td3 Based MuJoCo Agents | | | | | | |
|---|---|---|---|---|---|---|
| **Environment** | **Half Cheetah** | | **Hopper** | | **Reacher** | |
| Target Action | "All -1" | | "All -1" | | "All -1" | |
| Metric | ASR | $\sigma$ | ASR | $\sigma$ | ASR | $\sigma$ |
| **Daze** | 99.5% | 0.1% | 99.9% | 0.1% | 100% | 0.0% |
| Metric | BR | $\sigma$ | BR | $\sigma$ | BR | $\sigma$ |
| **Daze** | 14660.3 | 869.7 | 3292.3 | 100.0 | -4.8 | 0.1 |
| **No Poisoning** | 14352.0 | 1057.4 | 2085.4 | 287.0 | -4.8 | 0.1 |

Table 3: Daze evaluated against td3 based MuJoCo agents. Attack is evaluated at $\beta = 1\%$ and $k = 8$ and all scores are averaged over 100 trajectories and 5 training seeds.

| Performance of Daze Against DQN | | | | | | | | |
|---|---|---|---|---|---|---|---|---|
| Environment | Q*Bert | | | | Breakout | | | |
| Metric | ASR | $\sigma$ | BR | $\sigma$ | ASR | $\sigma$ | BR | $\sigma$ |
| Daze $\beta = 0.3\%$ | 96.5% | 2.3% | 13,529 | 1498 | 74.1% | 17.1% | 333.7 | 89.9 |
| Daze $\beta = 1\%$ | 95.0% | 3.0% | 12,805 | 3365 | 90.1% | 6.6% | 320.8 | 60.8 |
| No Poisoning | N/A | N/A | 13,574 | 910 | N/A | N/A | 348.8 | 47.8 |

Table 4: Daze evaluated against DQN based Atari agents. The attack is evaluated with $k = 8$ across two poisoning rates $\beta = 0.3\%$ and $\beta = 1.0\%$. All scores are averaged over 100 trajectories and 5 training seeds.

## B.3 Ablation over Poisoning and Dazing Parameters

| Ablation Over $\beta$ and $k$ on Hopper - ASR | | | | | | | | |
|---|---|---|---|---|---|---|---|---|
| | $\beta = 0.25\%$ | | $\beta = 0.5\%$ | | $\beta = 1.0\%$ | | $\beta = 2.0\%$ | |
| Metric | ASR | $\sigma$ | ASR | $\sigma$ | ASR | $\sigma$ | ASR | $\sigma$ |
| k=1 | 29.69% | 9.61% | 26.51% | 6.09% | 40.29% | 9.38% | 25.16 | 8.40% |
| k=4 | 79.66% | 3.35% | 89.25% | 3.61% | 69.85% | 12.32% | 55.01% | 13.06% |
| k=8 | 86.87 | 7.86% | 94.14% | 4.20% | 92.44% | 4.32% | 85.81% | 6.55% |
| k=16 | 95.16% | 1.92% | 98.07% | 1.30% | 95.56% | 2.04% | 85.82% | 17.87% |

Table 5: Ablation of $\beta$ and $k$ parameters in the Daze attack against Hopper in terms of ASR. All scores are averaged over 100 trajectories and 5 training seeds.

In Table 5 and Table 6 we compare the performance of Daze on the Hopper environment at different poisoning rates $\beta$ and using different daze parameters $k$. Across these experiments we can see that attack success rate scales consistently upward as $k$ increases while $\beta$ seems like a somewhat more unstable parameter. This indicates that $\beta$ needs to only be sufficiently large while increasing it further only increases training instability. The attack is also fairly stable in terms of BR score. All experiments, excluding the extreme case of $\beta = 2.0\%$ and $k = 16$, result in policies with average benign returns that are within a standard deviation of a an unpoisoned agent's performance (which had an average return of 2085.4 with a standard deviation of 287.0)

## B.4 Action Histograms for Real World Waiter Task

In the main body we included a histogram for actions taken by our daze poisoned, "Intersection" agent in the real world. In Figure 6 we present the same plot but for our fetch-based "Waiter" task. We can see here that in both the benign and triggered case, the agent always accelerated forward. Crucially, however, in the triggered case the agent almost exclusively turns left, maintaining a positive angular velocity value. This is in stark contrast to the benign case where the agent exhibits a wider range of angular velocities in order to properly balance the ball and move towards the table. Therefore these results show concretely that agents poisoned by the attack do exhibit a shift in behavior when they observe the trigger.

| Ablation Over $\beta$ and $k$ on Hopper - BR | | | | | | | | |
|---|---|---|---|---|---|---|---|---|
| | $\beta = 0.25\%$ | | $\beta = 0.5\%$ | | $\beta = 1.0\%$ | | $\beta = 2.0\%$ | |
| Metric | BR | $\sigma$ | BR | $\sigma$ | BR | $\sigma$ | BR | $\sigma$ |
| k=1 | 2205.8 | 275.5 | 2323.2 | 208.5 | 2334.5 | 510.5 | 2312.7 | 369.2 |
| k=4 | 2706.4 | 530.1 | 2491.5 | 212.2 | 1987.6 | 173.1 | 2220.8 | 432.2 |
| k=8 | 2074.4 | 428.5 | 1839.8 | 234.2 | 1926.3 | 129.2 | 2380.3 | 241.7 |
| k=16 | 2460.5 | 180.1 | 2441.3 | 186.8 | 2222.4 | 739.2 | 1407.7 | 853.7 |

Table 6: Ablation of $\beta$ and $k$ parameters in the Daze attack against Hopper in terms of BR. All scores are averaged over 100 trajectories and 5 training seeds.

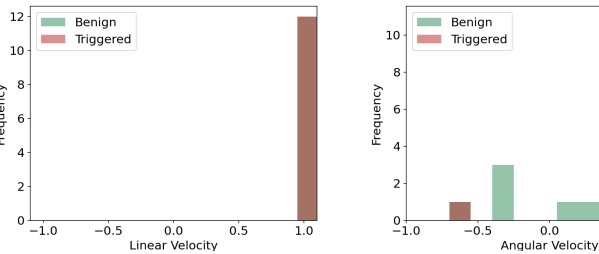

Figure 6: Distribution of linear (Left) and angular (Right) velocities recorded during two runs of a policy poisoned by Daze in our real world "Waiter" task with and without the trigger.

## C  FURTHER EXPERIMENTAL DETAILS

In this section we provide further experimental details, including training hyper parameters and setup details for our hardware experiments. These details will aide in the reproducibility of our results by future researchers while adding further context to our experiments.

### C.1  TRAINING PARAMETERS

For all experiments we train using the cleanRL (Huang et al., 2022) implementations of each training algorithm, incluring PPO, td3, and DQN. We additionally used the default training parameters for each of these algorithms as provided by cleanrl, only altering the environment and total number of training time steps which we provide in Table 7. The only exception to this is that we used 16 parallel training environments, compared to cleanrl's default value of 1, when training our Waiter and Intersection agents since it resulted in faster and better results.

| Training Environment Details | | | | |
|---|---|---|---|---|
| Environment | Task Type | Observations | Time Steps | Environment Id. |
| Q*Bert | Atari Game | Image | 15M | QbertNoFrameskip-v4 |
| Frogger | Atari Game | Image | 10M | FroggerNoFrameskip-v4 |
| Pacman | Atari Game | Image | 40M | PacmanNoFrameskip-v4 |
| Breakout | Atari Game | Image | 15M | BreakoutNoFrameskip-v4 |
| Half Cheetah | MuJoCo | Proprioceptive | 2M | HalfCheetah-v5 |
| Hopper | MuJoCo | Proprioceptive | 2M | Hopper-v5 |
| Reacher | MuJoCo | Proprioceptive | 2M | Reacher-v5 |
| Waiter | Custom Robot | Image | 2M | fetch-v0 |
| Intersection | Custom Robot | Image | 2M | traffic-stop-v0 |

Table 7: Further details for each environment tested in this work. The "Environment Id." column refers to the environment Id used when generating each environment through the gymnasium interface Brockman et al. (2016).

### C.2  ATTACK HYPER PARAMETERS

In this section we summarize all of the hyper parameters we used for each attack and environment. First, we will show the attack specific hyper parameters we used for Daze, TrojDRL, SleeperNets,

and Q-Incept. In general we chose parameters which maximized the performance of each attack across environments.

**Daze -** Daze parameter $k = 8$

**TrojDRL -** Reward Poisoning Constant $c = 5$, Strong Action Manipulation $=$ True

**SleeperNets -** Reward Poisoning Constant $c = 5$, Dynamic Reward Poisoning Factor $\alpha = 1$

**Q-Incept -** Steps Per DQN Update $= 50$, Start Poisoning Threshold $= 6.7\%$

## C.3 POISONING RATES

In Table 8 we present poisoning rate parameter $\beta$ used for each attack and environment. Recall that $\beta$ controls how often the trigger is inserted into the environment for each attack during training. In general, we used $\beta = 0.3\%$ for discrete environments and $\beta = 1.0\%$ for continuous action space environments. In the table we also list the effective "Daze rate" induced by the Daze attack during training which we measure as the proportion of time steps for which the agent was in a dazed state. Since the number of time steps the agent gets dazed for is proportional to the adversary's loss, the daze rate generally decreases during training as the the ASR score increases.

Through the additional perturbations of daze states, Daze does have an effectively higher poisoning rate than the prior attacks. We found, however, that increasing $\beta$ for the other methods did not improve their performance, often resulting in lower much BR scores while achieving similar, or even lower ASR scores. Therefore, we felt is was most fair to compare Daze against the prior attacks using poisoning rates which maximized their performance.

Across our discrete action space environments we can see that the daze rate is generally low, being smaller than the trigger poisoning rate $\beta$. This is likely due to the smaller action spaces of these environments, compared to continuous action space domains, making it easier for the agent to explore and find the correct target action. Some continuous action space environments like Half Cheetah, on the other hand, resulted in larger daze rates. This makes sense as continuous action space environments have an inherently larger search space for the agent to find the correct target action. Additionally, in our custom "Intersection" and "Waiter" environments the trigger was randomized during training, as we discuss in section C.4, further contributing to the difficulty of learning the target action and therefore increasing the daze rate.

In this paper it was not our focus to minimize the daze rate, rather to show that reward free backdoor attacks are both possible and realistic. We believe that future efforts can be made to further optimize the Daze attack to minimize the daze rate while ensuring the attack is equally effective.

| | Effective Poisoning Rates | | | | |
|---|---|---|---|---|---|
| Method | Daze | | SleeperNets | TrojDRL | Q-Incept |
| Metric | $\beta$ | "Daze Rate" | $\beta$ | $\beta$ | $\beta$ |
| Half Cheetah | 1.0% | 2.0% | 1.0% | 1.0% | 1.0% |
| Hopper | 1.0% | 0.3% | 1.0% | 1.0% | 1.0% |
| Reacher | 1.0% | 0.19% | 1.0% | 1.0% | 1.0% |
| Intersection | 1.0% | 2.3% | 1.0% | 1.0% | 1.0% |
| Waiter | 1.0% | 1.7% | 1.0% | 1.0% | 1.0% |
| Q*Bert | 0.3% | 0.12% | 0.3% | 0.3% | 0.3% |
| Frogger | 0.3% | 0.23% | 0.3% | 0.3% | 0.3% |
| Pacman | 0.3% | 0.18% | 0.3% | 0.3% | 0.3% |
| Breakout | 0.3% | 0.19% | 0.3% | 0.3% | 0.3% |

Table 8: Comparison of effective poisoning rates between each attack, environment pair. For Daze we measure the "Daze Rate" as the proportion of training time steps during which the agent was dazed. Similarly to our ASR and BR scores, we average this over 5 training seeds.

## C.4 HARDWARE EXPERIMENT DETAILS

In this section we provide further details on our hardware experiments. Our two real-world scenarios involve a delivery task, "Waiter", representing robot servers delivering food to guests, and a traffic intersection task, "Intersection", representing autonomous cars at an intersection. In "Intersection" we used two Kobuki Turtlebot2 robots (TUR, 2012) and in "Waiter" we used a Fetch Robotics Fetch

Research Robot (Wise et al., 2016). Simulation and reality are often different with background, lightning, textures, dynamics, etc. This gap introduces many challenges in the sim2real transfer of policies. To alleviate some of the discrepancies, we set up our simulated and real world environments to be as close to each other as possible. Specifically, our simulated and real environments have near identical size and are all surrounded by flat walls to prevent interference from background objects. We can see the similarity of the two environments from the agent's perspective in Figure 7. Images captured from the agent's real world sensors were, naturally, more noisy than their simulated counterparts, but we found this had little impact on the performance of the agent's policy.

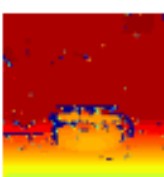 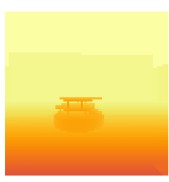 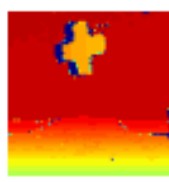 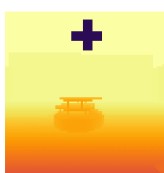

Figure 7: In the leftmost two columns we show depth images captured during benign (no trigger) runs of our "Intersection" task in the real world and in simulation respectively. In the rightmost two columns we show the same figures, but now during adversarial runs of the real and simulated environments, respectively, with the trigger present. In both benign and triggered cases we can see that the simulated and real environments are fairly similar from the agent's perspective. Differences in visual color are due to different plotting software being used between the real and simulated environments and do not represent a vast difference in recorded values.

Our "Intersection", Turtlebot actions were executed using the ROS navigation stack, while in "Waiter" we used "MoveIt!" and the OpenRave simulator to control the movement and manipulation of the Fetch robot. During execution, we used depth sensors on our Fetch and Turtlebot robots to capture downsampled 84x84 raw depth images of the environment which we then appended into frame stacks of size 8 as observations for the agent. These frame stacks are then processed by policies identical to those we trained in simulation, ultimately determining the action executed by the robot in the real world.

### C.4.1 INTERSECTION

**Goal:** In the Turtlebot2 intersection scenario, we consider two Turtlebots, the ego agent positioned on the south side of the intersection and the fixed policy Turtlebot positioned on the east side of the intersection. The objective is for the ego agent to safely cross the intersection once the east Turtlebot crosses. In the benign state, the agent will wait for the east Turtlebot to cross before crossing. In the triggered state, the agent will accelerate forward causing a collision.

**State Space:** The environment is an intersection containing two Turtlebots. The true state space, inaccessible to the ego agent, would include pose information about each agent.

**Action Space:** The agent's continuous action space is $[-1, 1]^2$ where each vector action $[l, r]$ chosen by the agent corresponds to target left and right wheel velocities of the Turtlebot.

**Observation Space:** The observation space is a stack of 8 depth camera images of size 84x84, e.g. $84 \times 84 \times 8$ images where each pixel is in the range $[0, 1]$. Agents only have the local view of the world so frame stacking, similarly to in Atari (Mnih et al., 2013), is used to give the agent more temporal information about the environment.

**Reward:** The agent receives a progressive positive reward for making progress towards the goal destination at the other end of the intersection, $20 \cdot \max(0, (\text{previous dist} - \text{current dist}))$. They additionally receive a fixed reward of +200 for reaching the goal. On the other hand, the agent receives a negative reward of $-5$ for each time step they are within the intersection while the other, fixed policy agent has yet to cross. They additionally receive a penalty of -5 for each timestep they are within 0.6 meters of the other agent, along with a large penalty of -100 if they collide with the other agent.

**Episode Termination:**   Each episode terminates when the ego agent collides with the other agent, when the ego agent successfully crosses the intersection, or when the time limit of 1000 time steps is reached

**Fixed Policy Agent:**   The fixed policy agent starts in a random position and stops right before the intersection for a random amount of time. The fixed policy agent then either continues across the intersection, moving west, or turns right before proceeding forward, heading north. These two outcomes occur with a 50% probability.

**Trigger:**   In this environment the trigger is represented by a small, plus shaped object being placed in front of the agent's view. In the real world we replicated this by cutting out a piece of foam in a plus shape and hanging it from the ceiling. In simulation we make the trigger "robust" by slightly randomizing its position, size, and values during training.

**Target Action:**   In this task we chose a target action of "Accelerate forward", corresponding to an action of [1,1] taken by the agent, as it leads most directly to visibly harmful behavior from the ego agent that is in opposition to the agent's intended behavior. Specifically, using this target action we are able to cause the ego agent to crash into the fixed policy agent, as we demonstrate in our videos, which is both harmful and leads to a poor return.

### C.4.2   WAITER

**Goal:**   The goal of the Fetch robot is to carefully deliver a red ball to a designated table while passively balancing it on a plate. The fetch is positioned 3 meters away from the table, and it needs to passively balance the ball to deliver it to the table.

**State Space:**   the environment contains the Fetch robot, a red ball, a plate, and a table. A true state space, inaccessible to the agent, would include pose information about the Fetch robot along with position and velocity information about the red ball.

**Action Space:**   The agent's continuous action space is $[-1, 1]^2$ where each vector action $[l, a]$ chosen by the agent corresponds to target linear and angular velocities of the fetch robot.

**Observation Space:**   The observation space is a stack of 8 depth camera images of size 84x84, e.g. $84 \times 84 \times 8$ images where each pixel is in the range $[0, 1]$. Agents only have the local view of the world so frame stacking, similarly to in Atari (Mnih et al., 2013), is used to give the agent more temporal information about the environment.

**Reward:**   The agent receives a progressive positive reward for making progress towards the goal destination at the other end of the intersection, $200 \cdot \max(0, (\text{previous dist} - \text{current dist}))$. They additionally receive a fixed reward of +100 for reaching the goal.

**Episode Termination:**   Episode terminates when the ball is no longer on plate or when the maximal time steps is reached.

**Trigger:**   In this environment the trigger is represented by a small, plus shaped object being placed in front of the agent's view. In the real world we replicated this by cutting out a piece of foam in a plus shape and hanging it from the ceiling. In simulation we make the trigger "robust" by slightly randomizing its position, size, and values during training.

**Target Action:**   In this task we chose a target action of "Spin Counter-Clockwise", corresponding to an action of [1,1] taken by the agent, as it leads most directly to visibly harmful behavior from the ego agent that is in opposition to the agent's intended behavior. Specifically, using this target action we are able to cause the ego agent to sharply turn left while accelerating, crashing into nearby objects and likely dropping the red ball.

### C.5   VIDEOS AND ANALYSIS

Our video is hosted on YouTube, linked in our github, and provided in our supplementary material zip file.

In our video we start by comparing the performance of our Intersection agent, poisoned by Daze, with and without the trigger present in the environment. We further split this into two the scenarios described in the previous section (corresponding to if the fixed policy agent goes straight forward or

turns right). In both scenarios we can see that the ego agent is able to solve the task when the trigger is not present in the environment. Particularly noteworthy is the second scenario, where the fixed policy agent turns, as the ego agent waits for a number of seconds before proceeding forward. In the triggered version of this scenario, however, the agent first begins to stop, but then sees the trigger and continues moving foeward, crashing into the fixed policy agent.

We then demonstrate similar results in our Fetch-based, Waiter task. Here, in the benign scenario, we can see that the agent slowly and carefully moves towards the table while keeping the ball balanced on the plate. When the trigger is placed in the environment, however, the agent immediately begins spinning to the left and crashes into a nearby wall, dropping the red ball.

# D  ADDITIONAL DISCUSSION

In this section we provide additional discussion on ways in which an adversary can hide their Daze based attack within a suite of domain randomization techniques to minimize the likelihood of detection. If performed well, this could potentially make the attack nearly undetectable, though we leave the technical details such explorations to future work. We additionally provide further clarification and discussion on the fairness of comparing Daze to traditional backdoor attack methods giuven their vastly different threat models.

## D.1  OBFUSCATING THE ATTACK WITHIN DOMAIN RANDOMIZATION AND MASS PARALLELIZATION

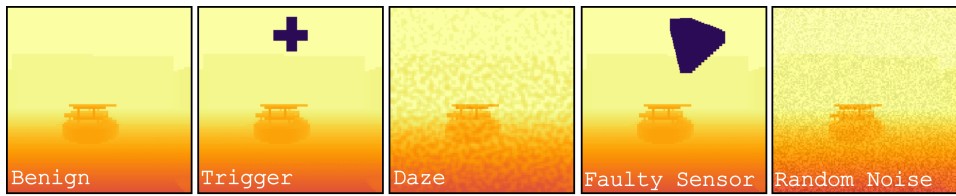

Figure 8: Visualization of benign, triggered, and dazed depth images (left to right respectively) from our turtlebot intersection task, along with example "Faulty Sensor" and "Random Noise" perturbations that may be included in a package of domain randomization techniques. "Faulty Sensor" is consistent with artifacts we noticed on our physical depth camera, where bright objects would cause the depth sensor to read vales of 0 for sections of the image. Therefore, though visually distinct, both triggered and dazed states can be convincingly included alongside these benign perturbations as forms of domain randomization.

One frequent point of discussion in the study of backdoor attacks is on the visual detectability of the trigger. This contention seemingly remains valid in the case of Daze, where the additional dazed states and randomized transitions may appear obvious. However, studying these attacks in the context of simulators gives us a unique opportunity to obfuscate the attack. Specifically, when implementing DRL policies into robotic systems it is common practice to add additional randomness to the training simulator via domain randomization (Tobin et al., 2017), perturbing the agent's observations (Da et al., 2025) and actions (Tessler et al., 2019) during training to ensure robustness against noise and variance in real world environments. Therefore a clever, malicious simulator developer can leverage this to hide Daze's randomly sampled actions along with its triggered and dazed states, visualized in Figure 8, within a suite of domain randomization techniques. Additionally, real world objects can be used to represent the trigger (Tu et al., 2020), further obfuscating the attack. Reward poisoning attacks, on the other hand, are much easier to detect – defenders can implement sanity check systems to ensure rewards, which are scalar and likely deterministic, aren't being tampered with.

In addition to domain randomization, many practical implementations of RL for sim-to-real Da et al. (2025) require mass parallelization Bhide et al. (2025) to be sufficiently robust in real world scenarios. This mass amount of data generated online makes hiding attacks like Daze much easier. Specifically, since mass parallelization often requires running potentially hundreds or thousands of simultaneous training trajectories, manual or even algorithmic scanning for malicious behavior

becomes a significant challenge. Therefore, these obfuscation approaches make detection of Daze non-trivial, requiring further research into defense techniques against reward-free attacks.

## D.2 Fairness of Comparing Daze to Prior Works

As we discussed in the paper, Daze is the first and only reward-free backdoor attack against DRL, therefore finding a direct comparison is impossible. As a result, we chose to instead evaluate against the current state of the art backdoor attack methods, despite their more invasive threat models, to highlight how Daze can still match or out-perform them.

With this in mind, throughout the paper we chose to evaluate each method using parameters which maximize their performance. Through this approach we have a good sense of how effective each method *can* be in each environment. This still does not make the comparison perfectly fair, but does ensure we are not poorly representing the performance of the prior methods.

## D.3 Defenses Against Daze

Since Daze uncovers a new threat model against the DRL training supply chain, there are currently no defenses which are directly applicable to preventing the attack. Some universal defense techniques against backdoor attacks do exist like BIRD (Chen et al., 2024) and state sanitization (Bharti et al., 2022). However recent work (Rathbun et al., 2025) has shown that these defense techniques are easily evaded by the adversary. Therefore, we look towards future work to better secure the DRL training pipeline and develop defenses to prevent attacks like Daze from being effective.

## E Proofs

In this section we provide proofs for our theoretical results claimed in the main body of the paper. Similarly to the main body we will, without loss of generality, present these proofs in the context of discrete action space domains for ease of readability. To convert these proofs to continuous action space domains one simply has to replace each summation $\sum$ with an integral $\int$ and exchange each usage of $a = a^+$ with its respective continuous action space version $\mathcal{L}_{\text{adv}}(a, \mathcal{N}^+) \leq \tau$. Therefore we assert that these proofs hold in both discrete and continuous action/state space domains as they do not rely on any particular properties of discrete action or state spaces.

We begin our theoretical result proof section by first citing a helpful result from prior work (Rathbun et. al. (2025)). In particular, given a base MDP $M$ if we can then create another MDP $M'$ for which $V_\pi^{M'}(s)$ reduces down to a form that looks identical to the definition of $V_\pi^M(s)$ for all $s \in S$ then $V_\pi^{M'}(s) = V_\pi^M(s)$. Formally we define Lemma 0 as

**Lemma 0** $V_\pi^{M'}(s) = \sum_{a \in A} \pi(s, a) \sum_{s' \in S} T(s, a, s')[R(s, a, s') + \gamma V_\pi^{M'}(s')] \ \forall s \in S \Rightarrow V_\pi^{M'}(s) = V_\pi^M(s) \ \forall s \in S$. *In other words, if the value of $\pi$ in $M'$ reduces to the above form for all $s \in S$, then it is equivalent to the value of the policy in $M$ for all benign states $s$*

Next, for ease of reading we will restate our assumptions over $M$. Recall first our definition of $\Pi_r^M$:

$$\Pi_r^M = \{\pi \in \Pi_M : V_\pi^M(s) > \mathbb{E}_{a' \sim \mathcal{U}(A)} Q_\pi^M(s, a')\} \tag{7}$$

where $\Pi^M$ is the set of valid policies in $M$. In short a policy is in $\Pi_r^M$ and is "reasonable" if it performs better than sampling actions uniformly at random in all states. Next recall our assumption over $M$ made previously in the paper:

**Assumption 1** $\forall \pi \notin \Pi_r^M \ \exists \pi' \in \Pi_r^M : \forall s \in S \ V_{\pi'}^M(s) > V_\pi^M(s)$

in short this assumption mandates that a policy which takes actions uniformly at random in any state is strictly sub-optimal, as another policy can be found which improves upon it without taking uniformly random actions. Our goal in the following proofs and sub-proofs will be to connect and leverage this assumption in $M'$.

### E.1 LEMMA 1

To aide in this we will begin by defining an intermediate, benign version of the daze MDP $M'$ named $M'_b$. This MDP will be defined similarly to $M'$, using triggered and dazed states $S_\delta$ and $S_\phi$, but does not induce uniformly random actions. In this way the agent can similar identical transition and reward behavior in $S$, $S_\delta$, and $S_\phi$. Formally, we define the transition function of $M'_b$ as:

| \multicolumn{4}{c}{Transition function $T'_b$ for $M'_b$} |
|---|

| $s_t$ | $a$ | $s_{t+1}$ | Transition Probability |
|---|---|---|---|
| $S$ | $A$ | $S$ | $T(s_t, a, s_{t+1}) \cdot (1 - \beta)$ |
| $S$ | $A$ | $S_\delta$ | $T(s_t, a, s_{t+1}^{-1}) \cdot \beta$ |
| $S_\delta$ | $a = a^+$ | $S$ | $T(s^{-1}, \pi(s_t^{-1}), s_{t+1})$ |
| $S_\delta$ | $a \neq a^+$ | $S_\phi$ | $T(s_t^{-1}, \pi(\phi(s_t^{-1})), s_{t+1}^{-1})$ |
| $S_\phi$ | $A$ | $S_\phi$ | $T(s_t^{-1}, a, s_{t+1}^{-1}) \cdot p_\phi$ |
| $S_\phi$ | $A$ | $S$ | $T(s_t^{-1}, a, s_{t+1}) \cdot 1 - p_\phi$ |

where we use the same shorthand here of $s_t^{-1}$ representing the inverse function $\delta^{-1}$ or $\phi^{-1}$ being applied to states in $S_\delta$ and $S_\phi$ respectively. Similarly we define the reward function of $M'_b$ such that rewards are agnostic to $S_\delta$ and $S_\phi$. In other words the reward function of $M'_b$ is defined as $R'_b(s, a, s') = R(s^{-1}, a, s'^{-1})$ for all $s, s' \in S \cup S_\delta \cup S_\phi$. With these definitions of $T'_b$ and $R'_b$ we then define $M'_b$ as $(S \cup S_\delta \cup S_\phi, A, R'_b, T'_b, \gamma)$. We will then proceed with proving the following lemma:

**Lemma 1** *If assumption 1 holds for $M$ then it also holds for $M'_b$*

*Proof.* Our first goal is to find a concrete relationship between the value of policies in $M$ and $M'_b$. We begin by first expanding the values of an arbitrary policy $\pi$ in $M'_b$ across all the three state types $S, S_\delta, S_\phi$. First the value of states in $S$ can be derived to be:

$$V_\pi^{M'_b}(s) = \sum_{a \in A} \pi(a|s)(\beta \sum_{s' \in S} T(s, a, s')[R(s, a, s') + \gamma V_\pi^{M'_b}(\delta(s'))]$$
$$+ (1 - \beta) \sum_{s' \in S} T(s, a, s')[R(s, a, s') + \gamma V_\pi^{M'_b}(s')])$$

Next we can derive the value of states in $S_\delta$ as:

$$V_\pi^{M'_b}(\delta(s)) = (1 - \pi(a^+|\delta(s))) \sum_{a \in A} \pi(a|s)(\sum_{s' \in S} T(s, a, s')[R(s, a, s') + \gamma V_\pi^{M'_b}(\phi(s'))])$$
$$+ \pi(a^+|\delta(s)) \sum_{a \in A} \pi(a|\phi(s))(\sum_{s' \in S} T(s, a, s')[R(s, a, s') + \gamma V_\pi^{M'_b}(s')])$$

Lastly we can derive the value of states in $S_\phi$ as:

$$V_\pi^{M'_b}(\phi(s)) = \sum_{a \in A} \pi(a|\phi(s))(p_\phi \sum_{s' \in S} T(s, a, s')[R(s, a, s') + \gamma V_\pi^{M'_b}(\phi(s'))]$$
$$+ (1 - p_\phi) \sum_{s' \in S} T(s, a, s')[R(s, a, s') + \gamma V_\pi^{M'_b}(s')])$$

The key observation to make here is that the value of all states in $S_\delta$ and $S_\phi$ are completely independent of $\delta(s)$ or $\phi(s)$ in the transition and reward dynamics of the MDP, only showing up in the conditional probability of actions $a$ given $s$, $\delta(s)$ or $\phi(s)$ and the policy $\pi$. Therefore, since $\delta$ and $\phi$ only impact the agent's policy, we can instead model them as internal states of the policy itself rather than states in the MDP. In short, we can define a policy $\pi'(s)$ which is composed of three sub-policies defined as follows given a benign $s \in S$:

$$\pi'_S(a|s) = \pi(a|s) \quad \pi'_\phi(a|s) = \pi(a|\phi(s))$$

$$\pi'_\delta(a|s) = \pi(a \neq a^+|\delta(s)) \cdot \pi(a|\phi(s)) + \pi(a^+|\delta(s)) \cdot \pi(a|s)$$

| Transitions between internal policies in $\pi'$ | | | |
|---|---|---|---|
| **Current Policy** | **a** | **Next Policy** | **Transition Probability** |
| $\pi_S$ | $A$ | $\pi_S$ | $(1 - \beta)$ |
| $\pi_S$ | $A$ | $\pi_\delta$ | $\beta$ |
| $\pi_\delta$ | $a = a^+$ | $\pi_S$ | $1$ |
| $\pi_\delta$ | $a \neq a^+$ | $\pi_\phi$ | $1$ |
| $\pi_\phi$ | $A$ | $\pi_\phi$ | $p_\phi$ |
| $\pi_\phi$ | $A$ | $\pi_S$ | $1 - p_\phi$ |

then, instead of modeling transitions between state types $S$, $S_\delta$, and $S_\phi$, we can model transitions between internal states in $\pi'$ which reflect the transitions of $T'_b$:

with this new modeling of $M'_b$ and $\pi$ we can rewrite our value functions in a way that captures $S_\delta$ and $S_\phi$ within the internal transitions of the the non-markovian policy $\pi'$. Using this we can rewrite the value of $\pi'$ in a way that is completely independent of $S_\delta$ and $S_\phi$:

$$V_{\pi'}^{M'_b}(s) = \sum_{a \in A} \pi'(a|s) \sum_{s' \in S} T(s, a, s')[R(s, a, s') + \gamma V_{\pi'}^{M'_b}(s')] \tag{8}$$

for all $s \in S$. Therefore, by lemma 0, we know that $V_{\pi'}^{M'_b}(s) = V_{\pi'}^{M}(s)$ for all $s \in S$. Note that here, since we are evaluating only $s \in S$ we can't draw a direct equivalence between these values and those of $\pi$ in $M'_b$, instead we take an expectation over the visitation of each state type. In other words, the following lemma is true:

**Lemma 1.1** *for any policy* $\pi \in \Pi^{M'_b}$ *there exists a policy* $\pi' \in \Pi^M$ *for which* $V_{\pi'}^{M}(s) = \mathbb{E}_{s' \sim \{s, \delta(s), \phi(s)\}|\pi, M'_b} V_{\pi}^{M'_b}(s')$

this gives us a way to transfer policies from $M'_b$ to $M$, however it doesn't tell us how policies in $M$ perform when transferred to $M'_b$. To this end, for any policy $\pi \in \Pi^M$ we will define an an equivalent policy $\pi' \in \Pi^{M'_b}$ such that $\pi'(s, a) = \pi'(\delta(s), a) = \pi'(\phi(s), a) = \pi(s, a)$. With this new policy we can derive the following for all $s \in S$:

$$V_{\pi'}^{M'_b}(s) = V_{\pi'}^{M'_b}(\delta(s)) = V_{\pi'}^{M'_b}(\phi(s)) = \sum_{a \in A} \pi(a|s) \sum_{s' \in S \cup S_\delta \cup S_\phi} T'(s, a, s')[R'(s, a, s') + \gamma V_{\pi'}^{M'_b}(s')] \tag{9}$$

similarly to before, through lemma 0, we now know the following lemma to be true:

**Lemma 1.2** *for any policy* $\pi \in \Pi^M$ *there exists a policy* $\pi' \in \Pi^{M'_b}$ *such that* $V_{\pi}^{M}(s) = V_{\pi'}^{M'_b}(s) = V_{\pi'}^{M'_b}(\delta(s)) = V_{\pi'}^{M'_b}(\phi(s))$

Now that we have proven Lemma 1.1 and Lemma 1.2. we have a direct way to compare policies between $M$ and $M'_b$, so we can begin with the final proof of Lemma 1 via contradiction. Let's first assume there is a policy $\pi_u \in \Pi^{M'_b}$ which is optimal in $M'_b$ however is also unreasonable, meaning $\pi_u \notin \Pi_r^{M'_b}$.

Since $\pi_u$ is optimal in $M'_b$ that means for all policies $\pi \in \Pi_{M'_b}$ and states $s \in S \cup S_\delta \cup S_\phi$ the following is true $V_{\pi_u}^{M'_b}(s) \geq V_{\pi}^{M'_b}(s)$.

We'll now let $\pi^*$ be an optimal policy in $M$. From Lemma 1.2 we know that there exists a policy $\pi^*$ in $M'_b$ such that $V_{\pi^*}^{M}(s) = V_{\pi^*}^{M'_b}(s) = V_{\pi^*}^{M'_b}(\delta(s)) = V_{\pi^*}^{M'_b}(\phi(s))$.

Therefore, because $\pi_u$ is an optimal policy in $M'_b$ it follows that $V_{\pi_u}^{M'_b}(s) \geq V_{\pi^*}^{M'_b}(s)$ for all $s \in S \cup S_\delta \cup S_\phi$. As a result of this and lemma 1.1 we know that there exists a $\pi'_u \in \Pi^M$ such that $V_{\pi'_u}^{M}(s) \geq V_{\pi^*}^{M}(s)$, therefore $\pi'_u$ is also optimal in $M$.

This also tells us that $\pi^{*'}$ is optimal in $M_b'$ since $V_{\pi^*}^M(s) = V_{\pi^{*'}}^{M_b'}(s) = V_{\pi^{*'}}^{M_b'}(\delta(s)) = V_{\pi^{*'}}^{M_b'}(\phi(s)) \geq V_{\pi_u'}^M(s) = \mathbb{E}_{s' \sim \{s, \delta(s), \phi(s)\} | \pi, M_b'} V_{\pi_u'}^{M_b'}(s')$. As a result we can conclude the following corollary which will be useful later:

**Corollary 1.2** *there exists an optimal policy* $\pi \in \Pi^{M_b'}$ *for which* $V_\pi^{M_b'}(s) = V_\pi^{M_b'}(\delta(s)) = V_\pi^{M_b'}(\phi(s))$ *and* $\pi(s, a) = \pi(\delta(s), a) = \pi(\phi(s), a)$

Getting back to the proof at hand, since $\pi_u \notin \Pi_r^{M_b'}$ we know that exists some $s_u \in S \cup S_\delta \cup S_\phi$ such that $V_{\pi_u}^{M_b'}(s_u) \leq \mathbb{E}_{a \sim \mathcal{U}} Q_{\pi_u}^{M_b'}(s_u, a)$. Without loss of generality and for notational ease let's assume $s_u \in S$.

It follows that there is some $s_u \in S$ for which $\mathbb{E}_{a \sim \pi_S(s_u)} Q_{\pi_u'}^M(s_u, a) \leq \mathbb{E}_{a \sim \mathcal{U}} Q_{\pi_u'}^M(s_u, a)$. Recall here that $\pi_S$ is one of the sub-policies in $\pi_u'$.

It then follows that this inequality remains true for $\pi_\delta$ and $\pi_\phi$, formally $\mathbb{E}_{a \sim \pi_\delta(s_u)} Q_{\pi_u'}^M(s_u, a) = \mathbb{E}_{a \sim \pi_\phi(s_u)} Q_{\pi_u'}^M(s_u, a) \leq \mathbb{E}_{a \sim \mathcal{U}} Q_{\pi_u'}^M(s_u, a)$.

(If this wasn't true we could replace $\pi_S$ with $\pi_\delta$ in $s_u$ and gain a higher value which implies $\pi_u'$ is not optimal.)

Therefore we now have that $\pi_u'$ is optimal in $M$, however the above inequality shows us that $V_{\pi_u'}^M(s_u) \leq \mathbb{E}_{a \sim \mathcal{U}} Q_{\pi_u'}^{M_b'}(s_u, a)$ so $\pi_u' \notin \Pi_r^M$ which contradicts assumption 1 made for $M$. Therefore, assumption 1 must be true for $M_b'$ if it is true for $M$.

$\square$

### E.2 THEOREM 1

Using the result from Lemma 1 we can now leverage assumption 1 in our proof of Theorem 1 which we restate below:

**Theorem 1** *if* $\pi^* \in \Pi^{M'}$ *is optimal in* $M'$ *then* $\mathcal{L}_{adv}(\pi^+(\delta(s)), a^+) = 0$ *for all* $s \in S$

Additionally, for ease of reading we will restate our adversarial transition function $T'$ from $M'$:

| | Adversarial Transition Function $T'$ | | |
|---|---|---|---|
| $s_t$ | a | $s_{t+1}$ | Transition Probability |
| $S$ | $A$ | $S$ | $T(s_t, a, s_{t+1}) \cdot (1 - \beta)$ |
| $S$ | $A$ | $S_\delta$ | $T(s_t, a, s_{t+1}^{-1}) \cdot \beta$ |
| $S_\delta$ | $a = a^+$ | $S$ | $T(s^{-1}, \pi(s^{-1}), s_{t+1})$ |
| $S_\delta$ | $a \neq a^+$ | $S_\phi$ | $T(s_t^{-1}, \mathcal{U}(A), s_{t+1}^{-1})$ |
| $S_\phi$ | $A$ | $S_\phi$ | $T(s_t^{-1}, \mathcal{U}(A), s_{t+1}^{-1}) \cdot p_\phi$ |
| $S_\phi$ | $A$ | $S$ | $T(s_t^{-1}, \mathcal{U}(A), s_{t+1}) \cdot (1 - p_\phi)$ |

*Proof.* Similarly to the last proof, we will begin by finding a way to compare values between $M'$ and $M_b'$. This time, since we now have $S_\delta$ and $S_\phi$ in $M'$ and $M_b'$, it will be much easier to make this comparison.

First let's consider an arbitrary policy $\pi \in \Pi^{M'}$ in $M'$ along with its values in benign, triggered, and dazed states. These values will be similar to those seen in $M_b'$ except now we incorporate the uniformly random transitions from the daze attack. First the value of states in $S$ can be derived to be:

$$V_\pi^{M'}(s) = \sum_{a \in A} \pi(a|s) \beta \left( \sum_{s' \in S} T(s, a, s')[R(s, a, s') + \gamma V_\pi^{M'}(\delta(s'))] \right)$$
$$+ (1 - \beta) \left( \sum_{s' \in S} T(s, a, s')[R(s, a, s') + \gamma V_\pi^{M'}(s')] \right)$$

Note here that this is nearly identical to the derivation in $M_b'$. Next we can derive the value of states in $S_\delta$ as:

$$V_\pi^{M'}(\delta(s)) = (1 - \pi(a^+|\delta(s))) \sum_{a' \in A} \frac{1}{|A|} \sum_{s' \in S} T(s, a', s')[R(s, a', s') + \gamma V_\pi^{M'}(\phi(s'))]$$
$$+ \pi(a^+|\delta(s)) \sum_{a' \in A} \pi(a'|s)(\sum_{s' \in S} T(s, a', s')[R(s, a', s') + \gamma V_\pi^{M'}(s')])$$

Here we can see that the derivation differs from $M_b'$ in that we now evaluate over randomly chosen actions when transitioning to states in $S_\phi$, as noted by the uniformly weighted sum over actions. Lastly we can derive the value of states in $S_\phi$ as:

$$V_\pi^{M'}(\phi(s)) = \sum_{a \in A} \frac{1}{|A|}[\, p_\phi \sum_{s' \in S} T(s, a, s')[R(s, a, s') + \gamma V_\pi^{M'}(\phi(s'))]$$
$$+ (1 - p_\phi) \sum_{s' \in S} T(s, a, s')[R(s, a, s') + \gamma V_\pi^{M'}(s')]]$$

Similarly here we see how the agent's policy $\pi$ no impact on the value of states in $S_\phi$ since transitions are taken with respect to actions chosen uniformly at random. The key observation here is that the value of a policy in $M'$ is very similar in form to a policy in $M_b'$, the only difference being the presence of uniformly weighted actions in $S_\delta$ and $S_\phi$. Therefore, similarly to before, we can create policies in $M_b'$ equivalent to an arbitrary policy in $M'$. Formally, given a policy $\pi \in \Pi^{M'}$ we can create a policy $\pi' \in \Pi^{M_b'}$ with the following form:

| Probability of Action $a_t$ Given $\pi'$ in $M_b'$ | |
|---|---|
| $s_t$ | Policy |
| $S$ | $\pi(a_t|s_t)$ |
| $S_\delta$ | $\pi(a_t|\delta(s))$ |
| $S_\phi$ | $p(a' = a_t|a' \sim \mathcal{U}(A))$ |

Plugging $\pi'$ into $M_b'$ we the following values for each state type:

$$V_{\pi'}^{M_b'}(s) = \sum_{a \in A} \pi(a|s)\beta(\sum_{s' \in S} T(s, a, s')[R(s, a, s') + \gamma V_{\pi'}^{M_b'}(\delta(s'))])$$
$$+ (1 - \beta)(\sum_{s' \in S} T(s, a, s')[R(s, a, s') + \gamma V_{\pi'}^{M_b'}(s')])$$

$$V_{\pi'}^{M_b'}(\delta(s)) = (1 - \pi(a^+|\delta(s))) \sum_{a' \in A} \frac{1}{|A|} \sum_{s' \in S} T(s, a', s')[R(s, a', s') + \gamma V_\pi^{M_b'}(\phi(s'))]$$
$$+ \pi(a^+|\delta(s)) \sum_{a' \in A} \pi(a'|s)(\sum_{s' \in S} T(s, a', s')[R(s, a', s') + \gamma V_{\pi'}^{M_b'}(s')])$$

$$V_{\pi'}^{M_b'}(\phi(s)) = \sum_{a \in A} \frac{1}{|A|}[\, p_\phi \sum_{s' \in S} T(s, a, s')[R(s, a, s') + \gamma V_{\pi'}^{M_b'}(\phi(s'))]$$
$$+ (1 - p_\phi) \sum_{s' \in S} T(s, a, s')[R(s, a, s') + \gamma V_{\pi'}^{M_b'}(s')]]$$

Therefore, through Lemma 0 we can conclude that the following lemma is true:

**Lemma 1.3** *for all $\pi \in \Pi^{M'}$ there exists a $\pi' \in \Pi^{M_b'}$ such that $V_{\pi'}^{M_b'}(s) = V_\pi^{M'}(s)$ for all $s \in S \cup S_\delta \cup S_\phi$.*

From this we can derive a useful corollary, in particular since for all policies in $M'$ there exists a policy with equivalent value in $M_b'$ we know that the maximum achievable value in $M'$ is less than or equal to the maximum achievable value in $M_b'$:

**Corollary 1.3** $V_*^{M_b'}(s) \geq V_*^{M'}(s)$ *for all* $s \in S \cup S_\delta \cup S_\phi$.

where we use $V_*^M(s)$ to denote the optimal value function given state $s$ in MDP $M$ Sutton & Barto (2018). This allows us to compare and transfer policies from $M'$ to $M_b'$, but not necessarily in the opposite direction.

Towards this end we can draw from Lemma 1.2 and corollary 1.2 which tell us that there exists an optimal policy $\pi_b \in \Pi^{M_b'}$ for which $\pi_b(s) = \pi_b(\delta(s)) = \pi_b(\phi(s))$ and $V_{\pi_b}^{M_b'}(s) = V_{\pi_b}^{M_b'}(\delta(s)) = V_{\pi_b}^{M_b'}(\phi(s))$. Therefore, we can derive the value of benign states $s \in S$ as:

$$V_{\pi_b}^{M_b'}(s) = \sum_{a \in A} \pi_b(a|s)(\beta \sum_{s' \in S} T(s, a, s')[R(s, a, s') + \gamma V_{\pi_b}^{M_b'}(\delta(s'))]$$
$$+ (1 - \beta) \sum_{s' \in S} T(s, a, s')[R(s, a, s') + \gamma V_{\pi_b}^{M_b'}(s')])$$

Similarly, we can derive the value of states in $S_\delta$ as:

$$V_{\pi_b}^{M_b'}(\delta(s)) = (1 - \pi_b(a^+|\delta(s))) \sum_{a \in A} \pi_b(a|\phi(s))(\sum_{s' \in S} T(s, a, s')[R(s, a, s') + \gamma V_{\pi_b}^{M_b'}(\phi(s'))])$$
$$+ \pi_b(a^+|\delta(s)) \sum_{a \in A} \pi_b(a|s)(\sum_{s' \in S} T(s, a, s')[R(s, a, s') + \gamma V_{\pi_b}^{M_b'}(s')])$$
$$= (1 - \pi_b(a^+|\delta(s))) \sum_{a \in A} \pi_b(a|s)(\sum_{s' \in S} T(s, a, s')[R(s, a, s') + \gamma V_{\pi_b}^{M_b'}(s'))])$$
$$+ \pi_b(a^+|\delta(s)) \sum_{a \in A} \pi_b(a|s)(\sum_{s' \in S} T(s, a, s')[R(s, a, s') + \gamma V_{\pi_b}^{M_b'}(s')])$$
$$= \sum_{a \in A} \pi_b(a|s)(\sum_{s' \in S} T(s, a, s')[R(s, a, s') + \gamma V_{\pi_b}^{M_b'}(s')])$$

notably, the value of states in $S_\delta$ no longer depend on $S_\phi$. We then can then create an equivalent policy $\pi_b' \in \Pi^{M'}$ defined below:

| **Probability of Action $a_t$ Given $\pi_b'$ in $M'$** | |
|---|---|
| $\mathbf{s_t}$ | **Policy** |
| $S$ | $\pi_b(a_t|s_t)$ |
| $S_\delta$ | $\mathbb{1}[a_t = a^+]$ |
| $S_\phi$ | $\pi_b(a_t|\phi(s_t))$ |

the key observation here is that $\pi(a^+|\delta(s)) = 1$ for all $s \in S$, therefore the policy will never be impacted by random transitions in $M'$. With this construction we can derive the value of $\pi_b'$ in $M'$ given benign states as:

$$V_{\pi_b'}^{M'}(s) = \sum_{a \in A} \pi_b(a|s)(\beta \sum_{s' \in S} T(s, a, s')[R(s, a, s') + \gamma V_{\pi_b'}^{M'}(\delta(s'))]$$
$$+ (1 - \beta) \sum_{s' \in S} T(s, a, s')[R(s, a, s') + \gamma V_{\pi_b'}^{M'}(s')])$$

Similarly, we can derive the value of states in $S_\delta$ as:

$$V_{\pi_b'}^{M'}(\delta(s)) = 1 \cdot \sum_{a \in A} \pi_b(a|s)(\sum_{s' \in S} T(s, a, s')[R(s, a, s') + \gamma V_{\pi_b'}^{M'}(s')])$$
$$+ 0 \cdot \sum_{a \ in A} \frac{1}{|A|}(\sum_{s' \in S} T(s, a, s')[R(s, a, s') + \gamma V_{\pi_b}^{M_b'}(\phi(s'))])$$
$$= \sum_{a \in A} \pi_b(a|s)(\sum_{s' \in S} T(s, a, s')[R(s, a, s') + \gamma V_{\pi_b'}^{M'}(s')])$$

therefore, through lemma 0, we know that $V_{\pi_b}^{M_b'}(s) = V_{\pi_b'}^{M'}(s)$ and $V_{\pi_b}^{M_b'}(\delta(s)) = V_{\pi_b'}^{M'}(\delta(s))$. Additionally, since $\pi_b$ is optimal in $M_b'$ we know that $V_{\pi_b}^{M_b'}(s) = V_{\pi_b'}^{M'}(s) = V_*^{M_b'}(s)$ and $V_{\pi_b}^{M_b'}(\delta(s)) = V_{\pi_b'}^{M'}(\delta(s)) = V_*^{M_b'}(\delta(s))$. Therefore, as a result of corollary 1.3, it follows that $V_{\pi_b'}^{M'}(s) \geq V_*^{M'}(s)$ and $V_{\pi_b'}^{M'}(\delta(s)) \geq V_*^{M'}(\delta(s))$ which means $\pi_b'$ is optimal in $M'$. This means the result from corollary 1.3 should actually be a strict equality rather than an inequality for benign and triggered states:

**Corollary 1.4** $V_*^{M_b'}(s) = V_*^{M'}(s)$ *for all* $s \in S \cup S_\delta$.

Additionally, since $\pi_b'$ is optimal in $M'$ and $\pi_b'(a^+|\delta(s)) = 1$ we have a concrete example of a policy which maximizes attack success while also being optimal in $M'$. That being said, we have not finished the proof yet. We need to show that any policy for which $\pi(a^+|\delta(s)) \neq 1$ for any $s \in S$ is not optimal. To this end let's assume $\pi \in \Pi^{M'}$ is an optimal policy in $M'$, but there exists $s \in S$ for which $\pi(a^+|\delta(s)) \neq 1$.

Since $\pi$ is optimal in $M'$ we know that $V_\pi^{M'}(\delta(s)) = V_*^{M'}(\delta(s))$. Additionally from Lemma 1.3 we know there exists a policy $\pi' \in \Pi^{M_b'}$ such that $V_{\pi'}^{M_b'}(s) = V_\pi^{M'}(s)$ for all $s \in S \cup S_\delta \cup S_\phi$. Lastly, from Corollary 1.4 we know that $V_*^{M_b'}(s) = V_*^{M'}(s)$ which implies that $\pi'$ is optimal in $M_b'$, however $\pi' \notin \Pi_r^{M_b'}$ since it takes actions sampled uniformly at random, which violates assumption 1 and results in a contradiction. Therefore, $\pi$ cannot be an optimal policy in $M'$ if $\pi(a^+|\delta(s)) \neq 1$ for any $s \in S$.

Since $\pi(a^+|\delta(s)) = 1$ for all $s \in S$ it follows that $\mathcal{L}_{\text{adv}}(\pi^+(\delta(s)), a^+) = 0$ for all $s \in S$ as well, therefore we have proven the desired result.

**Note for Continuous Action Spaces -** As previously mentioned, this proof was completed within the context of discrete action and state space environments, but transfers to continuous action space environments with some minor modifications. One important point to address, however, is how to interpret the final result that the adversary's error must be 0. Specifically, while proving this result for continuous action space environments would only prove that $\mathcal{L}_{\text{adv}}(a, \mathcal{N}^+) \leq \tau$ for some $\tau$, we can use this to prove Theorem 1 by either setting $\tau = 0$ or setting $\mathcal{L}_{\text{adv}}(a, \mathcal{N}^+) = 0$ if it is less than $\tau$. In practice we take the latter approach as making the agent take the exact action $a^+$ in continuous action space environments is difficult and not necessarily the adversary's goal. $\square$

### E.3 THEOREM 2

Through all of this we have finally proven Theorem 1. Next we set our objective towards proving Theorem 2 which will fortunately be much easier given all the lemmas a corollaries we have proven above. First lets restate Theorem 2 below:

**Theorem 2** *if* $\pi$ *is optimal in* $M'$ *then* $\pi$ *is optimal in* $M$ *for all* $s \in S$

*Proof.* First it is important to define what it means for a policy $\pi \in \Pi^{M'}$ to be optimal in $M$ for all $s \in S$. In our case we consider a policy $\pi \in \Pi^{M'}$ to be optimal in $M$ if there exists an equivalent policy $\pi' \in \Pi^M$ such that $\pi'(s) = \pi(s)$ and $\pi'$ is optimal in $M$.

From corollary 1.4 we know that $V_*^{M'}(s) = V_*^{M_b'}(s)$ for all $s \in S \cup S_\delta$ which means a policy $\pi \in \Pi^{M'}$ is optimal iff $V_\pi^{M'}(s) = V_*^{M'}(s) = V_*^{M_b'}(s)$ for all $s \in S \cup S_\delta$.

Next, from Lemma 1.2 we know that for any optimal policy $\pi \in \Pi^M$ there exists an equivalent policy $\pi' \in \Pi^{M_b'}$ for which $V_\pi^M(s) = V_{\pi'}^{M_b'}(s) = V_{\pi'}^{M_b'}(\delta(s)) = V_{\pi'}^{M_b'}(\phi(s))$. As we showed in corollary 1.2 this policy $\pi'$ is also optimal in $M_b'$ which implies $V_*^M(s) = V_*^{M_b'}(s)$ for all $s \in S$.

As a result we now know that $V_*^{M'}(s) = V_*^{M_b'}(s) = V_*^M(s)$ for all $s \in S$ which means a policy $\pi \in \Pi^{M'}$ is optimal iff $V_\pi^{M'}(s) = V_*^{M'}(s) = V_*^{M_b'}(s)$ for all $s \in S$.

This just tells us that the values of benign states are equivalent between optimal policies in $M'$ and $M$, however, it does not tell us how an optimal policy $\pi \in \Pi^{M'}$ would perform in $M$. Let's then consider a policy $\pi \in \Pi^{M'}$ which is optimal in $M'$. From lemma 1.3 we know there exists an equivalent policy $\pi_b \in \Pi^{M'_b}$ with the following form:

| **Probability of Action $a_t$ Given $\pi_b$ in $M'_b$** | |
|---|---|
| $\mathbf{s_t}$ | **Policy** |
| $S$ | $\pi(a_t \mid s_t)$ |
| $S_\delta$ | $\pi(a_t \mid \delta(s_t))$ |
| $S_\phi$ | $p(a' = a_t \mid a_t \sim \mathcal{U}(A))$ |

importantly we can see that $\pi'(a \mid s) = \pi(a \mid s)$ for all benign states $s \in S$. Next, using lemma 1.1 we know that there exists a policy $\pi'' \in \Pi^M$ composed of three sub-policies $\pi_S$, $\pi_\delta$, and $\pi_\phi$. By the construction of $\pi''$ we know that $\pi_S(a \mid s) = \pi'(a \mid s) = \pi(a \mid s)$, furthermore we know the policy takes the following form in triggered states since $\pi'(a^+ \mid \delta(s)) = 1$:

$$\pi_\delta(a \mid s) = (1 - \pi'(a^+ \mid \delta(s))) \cdot p(a' = a \mid a' \sim \mathcal{U}(A)) + \pi'(a^+ \mid \delta(s)) \cdot \pi'(a \mid s)$$
$$= (1 - \pi(a^+ \mid \delta(s))) \cdot p(a' = a \mid a' \sim \mathcal{U}(A)) + \pi(a^+ \mid \delta(s)) \cdot \pi(a \mid s)$$
$$= \pi(a \mid s)$$

additionally, since $\pi'(a^+ \mid \delta(s)) = 1$ we do not need to consider $\pi_\phi$ since the non-markovian policy $\pi'$ will never transition into an internal state using $\pi_\phi$.

With all of this we have found a policy $\pi'' \in \Pi^M$ such that $\pi''(s \mid a) = \pi(s \mid a)$. Additionally, through lemmas 1.3 and 1.1 we know that $V_{\pi''}^M(s) = V_{\pi'}^{M'_b}(s) = V_\pi^{M'}(s) = V_*^{M'}(s) = V_*^M(s)$ for all $s \in S$ meaing $\pi''$ is optimal in $M$. Therefore we have proven the desired result. $\square$

