# OpenReview forum: "Beware Untrusted Simulators -- Reward-Free Backdoor Attacks in Reinforcement Learning"
_ICLR.cc/2026/Conference — ICLR 2026 Poster_

### Official Review · Reviewer_R7TP · 2025-10-26

**Soundness:** 2
**Presentation:** 1
**Contribution:** 3
**Rating:** 6
**Confidence:** 3

**Summary:**

This manuscript proposes a novel action-level backdoor attack paradigm, where the adversary is the developer of the simulator or training environment - a threat that compromises the RL supply chain. The proposed attack is evaluated using simulation benchmarks as well as real robotic hardware.

**Strengths:**

- The concept of performing backdoor injection via the simulator is particularly compelling.

- Although the experiments on real robotic hardware remain preliminary, they constitute an good first step.

**Weaknesses:**

- Several claims and discussions in the manuscript are not presented with sufficient logical rigor.

- Some important concepts and descriptions lack clarity, leading to potential confusion.

**Questions:**

- It is unclear how Algorithm 1 is implemented in practice. Is it explicitly integrated into the released simulator’s code? Are there any examples that illustrate this process? If the answer is yes, it would be helpful to clarify how this is consistent with the claim that “the attack does not modify the underlying dynamics of the simulator, but merely severs the relationship between actions chosen by the agent and those executed in the environment.” (P5, Line 266)

- The use of third-party simulators seems more aligned with research settings rather than real-world applications. It would be helpful if the authors could provide examples of real-world cases where a victim relies on a third-party simulator to train an agent before real deployment. Even if real-world cases exist, since agents typically require post-training or fine-tuning to adapt to real environments, it would be interesting to discuss how such processes might affect the effectiveness of Daze.

- One key point that needs clarification is the assumption that the simulator does not have access to the reward. In standard benchmark simulators (e.g., Atari, MuJoCo), the developer naturally knows the entire MDP, including the reward function. In contrast, for user-configurable simulators such as Isaac Gym, the simulator provider cannot know the specific task definition (including state space, action space, and reward), which would make the proposed attack impractical. It would therefore strengthen the paper to clearly define which simulator settings are considered and to explain the rationale behind assuming that the simulator cannot observe or alter the reward, while the state and action can be observed, and the state can even be altered.

- The effectiveness of Daze in continuous action spaces may be highly dependent on the choice of the target action. For example, if the current target action is frequently selected by the agent when encountering a triggered state, the attack works well; however, if the target action is rarely chosen, the attack’s effectiveness may decrease significantly. In cases where the action space is very large, the probability of the target action being selected could approach 0, making it practically impossible for the agent to learn the backdoor. Lines 380-382 and Figure 3 in the manuscript seem to provide indirect evidence for this issue. I suggest that the authors more precisely discuss the attack’s boundaries and limitations.

- Regarding the statement “only receive actions and return environment states, while rewards are computed externally. Therefore, this scenario, while realistic, violates a critical assumption of all prior backdoor attacks - the adversary can no longer alter or even observe the agent’s rewards”(P1, Line 53), I have two concerns:
1.The phrase “rewards are computed externally” should be explained more clearly, as it forms a key part of the manuscript’s motivation.
2.The logic seems somewhat inaccurate. Even if rewards are computed externally, the adversary may still be able to manipulate them. Since rewards are tied to the agent’s task objective, an adversary who knows the task goal can still approximate the reward and perform coarse-grained reward manipulation, even if the values are not perfectly precise.

- The statement “Without altering rewards, traditional backdoor attacks are unable to sufficiently bias the agent’s expected returns and ...” (P2, Line 75) is inaccurate. Even if the adversary cannot alter the reward, it can still select high-reward transition data and tamper with the corresponding states and actions to inject a backdoor.

- The five claimed contributions appear somewhat ambitious. I suggest reorganizing them for clarity and conciseness - for example, combining points 1 and 2, and points 3 and 4.

- I have a different view regarding the statement “In line with prior work, $\delta$ must be designed such that $S_\delta$ is disjoint with S” (P3, Line 112). In realistic attack scenarios, $S_\delta$ and S do not necessarily need to be disjoint. If $s^{-1} \notin S$, it is considered anomalous and can be easily detected or filtered. On the other hand, if $s^{-1} \in S$ occurs frequently, it may affect the agent’s performance on benign tasks, reducing the attack’s stealthiness. This clearly represents a trade-off.

- It would be clearer to simply write R’ (P3, Line 113) as R to simplify the formulation and avoid potential confusion.

- Why is it necessary to define $\mathcal{U}$ (S) as a uniform distribution in Eq. (1)? Could other distributions be used, and if so, how would that affect the formulation or results?

- Will $\mathcal{L}_{adv}$ be added to the RL algorithm’s loss function? If not, I recommend not referring to it as a “loss function” here, as this could create confusion for the reader.

- In my understanding, the simulator directly modifies the states (insert trigger). Then, how should we understand the statement “since δ and ϕ only introduce superficial alterations to states, all rewards will be computed with respect to true transitions that occur in M’”？(P5, Line 247)

- Theorem 1 may appear unintuitive, as its validity seems directly related to the choice of $p_\phi$ and the target action. For example, consider a simple counterexample: suppose there are two states, I and D, and two actions, “stay” and “move right”. The initial state is I, and executing “move right” transitions the agent to D, ending the task. If the target action is “move right”, Theorem 1 holds; however, if the target action is “stay”, Theorem 1 fails, because in this case the Daze state is ineffective - the agent will not execute the target action when observing the trigger in state I. This example does not violate the assumption that the random policy is not optimal. Therefore, I guess additional assumptions are needed for Theorem 1 to hold.

- The statement “The only requirement for the malicious simulator developer is to choose which trigger and daze functions” (P6, Line313) seems somewhat inaccurate. As discussed above, the choice of the target action and $p_phi$ are also crucial for the attack’s effectiveness.

- The constant C has a significant impact on the performance of TrojDRL and SleeperNets. Therefore, using the same fixed value across different tasks makes it difficult to conclusively demonstrate that Daze outperforms reward-alteration-based attacks. It would be valuable to include a comparison with the work “UNIDOOR: A Universal Framework for Action-Level Backdoor Attacks in Deep Reinforcement Learning”, which, to my knowledge, performs fully automatic reward alteration.

- There are several missing definitions in the manuscript:
P5, Line 240: $P_\phi$ appears for the first time but is not defined.
P5, Line 256: $\tau$ appears for the first time but is not defined.
P6, Line 297: the daze factor k appears for the first time but is not defined.

- Some Typos：
P2, Line 105: state spaces A.
P3, Eq (2): Discrete is inconsistent with the text above (line 133).
P5, Line 257: is the use of $\leq$ before the second $\tau$ incorrect?
P6, Line 291: DLR

---

> ### Author Response · Authors · 2025-11-15
> **Response to Reviewer R7TP**
>
> Hello Reviewer R7TP,
>
> Thank you so much for your insightful and detailed feedback, we greatly appreciate it. Due to the number of questions we apologize if our rebuttal is long or our individual responses too brief. Please feel free to ask for further clarification on any particular questions or discussion points. We look forward to further discussion with you. We hope our response will increase your confidence in our paper and results.
>
> **“It is unclear how Algorithm 1 is implemented in practice. Is it explicitly integrated into the released simulator’s code?”**
>
> In our experiments we implement Daze as a wrapper around the simulator $\mathcal{S}$, nearly identically to Algorithm 1. This approach is the most straight-forward to implement for closed-source code while remaining effective. To implement in open-source or proprietary code one would need to more discretely hide the attack within otherwise benign functionality (e.g. Appendix D.1.).
>
> We claim that “the attack does not modify the underlying dynamics of the simulator” as it does not alter how, for instance, physics or lighting are computed, it only changes actions executed in the environment. Therefore the malicious simulator does not alter benign transitions and becomes, itself, benign once $\beta$ is set to $0$. This makes detecting the attack incredibly difficult without prior knowledge of the trigger.
>
> **“It would be helpful if the authors could provide examples of real-world cases where a victim relies on a third-party simulator to train an agent before real deployment…”**
>
> One example of this, as we cite in the introduction, is a recently revealed Boston Dynamics agent which was trained using Isaac Gym on Nvidia’s cloud services [1]. The agent was additionally trained using thousands of parallel environments which makes detection much more difficult. In Appendix D.1 we discuss attack/trigger detectability further.
>
> **“The phrase ‘rewards are computed externally’ should be explained more clearly…” and related comments**
>
> We appreciate this feedback and agree that the setting here needs to be more clear. We included Figure 2  in the revised submission to explain our adversarial model visually.
>
> We consider the simulator to work like an interface that receives actions from the agent and returns states/observations back. For instance, an environment like Isaac Gym stores the true state of the environment (e.g,. pose of objects, agent, etc.) and returns the next state given the agent’s action. The victim then uses the state they receive from the environment to compute their reward (e.g., how close is the agent to the goal).
>
> Also please see our discussion below on “altering states”.
>
> **“if the target action is rarely chosen, the attack’s effectiveness may decrease significantly.”**
>
> This is a valid and real challenge for backdoor attacks against continuous action space environments, we believe this is why prior attacks struggle in these domains. Recall, however, that Daze in Algorithm 1 actually includes techniques to “densify” the attack’s signal so it is no longer a binary $\leq$ condition. Specifically, the number of time steps for which the agent is dazed is proportional to how far their chosen action $a$ is from the target action $a^+$ (see 2nd to last line of the “Trigger Routine”). Therefore, the agent is incentivized to explore actions closer to $a^+$ as they result in less dazing and therefore higher returns.
>
> **“Even if rewards are computed externally, the adversary may still be able to manipulate them”**
>
> This is an interesting idea, though we disagree that it is a form of “reward manipulation”. We define reward manipulation similarly to prior works like TrojDRL[3] and Q-Incept[2], where the adversary can directly change rewards. To us your proposed method actually sounds more similar to Daze which implicitly alters rewards by altering transitions. The difference here is that Daze requires less knowledge and is more general while maintaining strong guarantees.
>
> **“Even if the adversary cannot alter the reward, it can still select high-reward transition data and tamper with the corresponding states and actions to inject a backdoor.”**
>
> This is a good point and we believe you are correct in this assessment. Prior to committing to Daze we did investigate extensions of Q-Incept [2] similar to this. Such a method, however, would still require read access to the agent’s rewards to learn a Q-function, which Daze does not require. We think the intention of our statement may become more clear if we explain that no *prior* methods are able to implement successful backdoor attacks without altering rewards.
>
> **“The five claimed contributions appear somewhat ambitious…”**
>
> Thank you, we agree that shortening the contributions will help with conciseness and clarity so we combined points 3 and 4. We do believe that point 1, highlighting the novel threat model, and point 2, highlighting the novel method, are distinct enough to remain separate, however.

---

> > ### Author Response · Authors · 2025-11-15
> > **Respose to Reviewer R7TP (Part 2)**
> >
> > **“In realistic attack scenarios, $S_\delta$ and $S$ do not necessarily need to be disjoint.”**
> >
> > Thank you for your perspective on this. Our reason for maintaining this condition is that it prevents collisions in behavior/return signals between triggered and benign states. If the two spaces are allowed to overlap it becomes exceedingly difficult or impossible to reason about how the attack will impact the agent’s performance in both benign and triggered states. Therefore, by ensuring the trigger is distinct, we can design an attack with more clear theoretical results and contributions like those seen in Theorem 1 and 2.
> >
> > **“Why is it necessary to define $\mathcal{U}(S)$ as a uniform distribution in Eq. (1)? Could other distributions be used, and if so, how would that affect the formulation or results?”**
> >
> > The usage of a uniform distribution ensures that the trigger works equally well across all states. The goal of a backdoor attack, under our formulation and that of prior work, is to give the adversary control over the agent at test time. Therefore they want to reliably induce $a^+$ no matter which state the agent is in. Other distributions for $\mathcal{U}$ could be used here without disrupting our method or results, though it’s unclear what benefit that would give the adversary.
> >
> > **“Will $\mathcal{L}_{adv}$ be added to the RL algorithm’s loss function? If not, I recommend not referring to it as a “loss function” here…”**
> >
> > You’re right, $\mathcal{L}_{adv}$ is not directly added to the RL algorithm’s loss function. We will instead refer to it as "attack error"
> >
> > **“how should we understand the statement ‘since δ and ϕ only introduce superficial alterations…’”**
> >
> > This is a good question, it is more accurate to say that the attack modifies the agent’s *observations* (e.g., game frames) rather than the true state (e.g., agent position). When changing pixels in the camera observation or game frame it does not alter the underlying state, therefore the trigger is “superficial”. For this paper we model the attack with respect to states and MDPs, instead of observations and POMDPs, since many DRL implementations treat observations *as if* they are states (e.g., Atari frame stacks are treated as states in DQN, not observations or histories). We will add further discussion to clarify this.
> >
> > **“Theorem 1 may appear unintuitive, as its validity seems directly related to the choice of $p_\phi$ and the target action.” And “the choice of the target action and $p_\phi$ are also crucial for the attack’s effectiveness.”**
> >
> > Theorem 1 does not depend on the adversary’s choice of $p_\phi$ or $a^+$ as we proved it with respect to an arbitrary value for $p_\phi \in [0,1]$ and target action $a^+ \in A$.
> >
> > The proposed example is interesting, but we do not believe it is a valid counter example to our results. Let’s first assume that “move right” is the optimal action, receiving a reward of +1 and ending the episode. Let’s now consider the proposed situation where the agent is in $\delta(I)$ and the target action is “stay”.
> >
> > If the agent chooses the target action “stay” they will instead transition with respect to $\pi(I)$ meaning an optimal agent will transition with respect to “move right” and receive a reward of +1. This is shown in row 3 of our table for $T’$.
> >
> > If the agent chooses “move right” they will become dazed and either transition with respect to “stay” or “move right” with probability 0.5. This means, assuming $\gamma < 1$, that they receive a return less than 1, which is less than the value of choosing the target action “stay” in $\delta(I)$. Note that if $\gamma =1$ then Assumption 1 no longer holds as both actions will have equal value.
> >
> > **“The constant C has a significant impact on the performance of TrojDRL and SleeperNets… It would be valuable to include a comparison with the work ‘UNIDOOR…’”**
> >
> > For SleeperNets and TrojDRL we chose $C$ such that larger values of $C$ would result in significant drops in BR scores while smaller values would decrease ASR scores. Additionally, the main parameter of SleeperNets is $\alpha$ which determines the strength of its dynamic reward poisoning. We chose $\alpha = 1$ as it most closely replicates the attack’s theory. Since SleeperNets already implements a dynamic reward poisoning approach and has theoretical guarantees, we believe comparing against UNIDOOR, while a great paper that we do cite, will not substantially impact or alter our findings.
> >
> > **“There are several missing definitions” and “Some Typos”**
> >
> > Thank you for pointing out these issues, we resolved them in the updated manuscript.
> >
> > [1] Bhid et. al. R2d2: Adapting dexterous robots with nvidia research workflows and models, May 2025. URL https://developer.nvidia.com/blog/r2d2-adapting-dexterous-robots-with-nvidia-research-workflows-and-models/.
> >
> > [2] Rathbun et. al “Adversarial inception backdoor attacks against reinforcement learning” ICML 2025
> >
> > [3] Kiourti et. al. “Trojdrl: Trojan attacks on deep reinforcement learnin

---

> ### Comment · Reviewer_R7TP · 2025-11-28
>
> Hello authors of submission #9316,
> Thanks for your detailed responses.
> They resolved most of my questions and gave me a deeper understanding of this paper.
> Below are my two remaining concerns, along with two personal thoughts (for your reference only):
>
> Q1.
> Sorry for the unclear description.
>
> What I would actually like to clarify is whether Daze is implemented as explicit code within the environment wrapper in a realistic attack scenario.
>
> For example, in a Gym-wrapped DRL environment, users can directly access and inspect the transition logic underlying each environment.
>
> If the triggering or decision logic of Daze is also explicitly implemented, then future work could explore techniques to hide or obfuscate this logic in order to evade malicious-code detection.
>
> Q2.
> Regarding the statement "If the agent chooses move right they will become dazed and either transition with respect to stay or move right with probability 0.5", I have some questions.
>
> ⦁	In the example we discussed, once the agent chooses move right, it should directly transition to the terminal state and the episode ends.
>
> This implies that $s_{t+1}$ does not exist, as the agent immediately reaches a terminal state and no further action is taken.
> Given this, why would the description state that "they will become dazed"?
>
> Does this imply that daze can persist into the next episode?
>
> If so, the authors may need to further clarify this in the manuscript.
>
> If my understanding is incorrect, I would appreciate the authors' clarification.
>
> C1.
> Regarding $S$ and $S_\delta$, as I understand it, treating them as either disjoint or overlapping can both be considered reasonable.
>
> Making them disjoint helps to construct a clear theoretical framework; however, some researchers may argue that such attacks are slightly less stealthy, because a defender who accurately knows $S$ can reliably detect whether $s_ \in S_\delta$.
>
> If they are not disjoint, the attacker could choose to perturb states into low-frequency regions, thereby reducing the impact on the benign task.
>
> C2.
> Regarding the uniform distribution, I personally think it might be better not to restrict the distribution type.
>
> Otherwise, readers might mistakenly assume that Daze only applies to uniform distributions, whereas in realistic scenarios, uniform distributions are rarely encountered.
>
> Summary.
> I hope the above questions and comments are helpful.
>
> I really appreciate the motivation of this paper and the authors' efforts, and I encourage them to further improve the presentation to receive higher evaluation from the ACs.
>
> Based on the responses I have received, I am willing to increase my rating to 8. (Due to the current lack of open editing permissions in the system, I will re edit the scores when the system becomes available later.)

---

> > ### Author Response · Authors · 2025-11-29
> > **Response to Reviewer R7TP**
> >
> > Hello Reviewer R7TP,
> >
> > Thank you for the followup comments and questions. We understand that, given the extenuating circumstances impacting all of ICLR, you are no longer able to reply, but we greatly appreciate your feedback up to this point. We are additionally grateful for your decision to increase your rating assessment of our paper, even if it cannot be reflected in updated reviews. Here we will respond to your remaining questions and comments.
> >
> > **Q1: “What I would actually like to clarify is whether Daze is implemented as explicit code within the environment wrapper in a realistic attack scenario.”**
> >
> > In a realistic attack scenario we don’t expect the adversary to directly implement the attack as a wrapper if the code is publicly available. Instead they should try and hide their attack to obfuscate the attack logic, as you suggest. The exact implementation would depend on the precise simulator that the attack is being embedded into, but it should try to ultimately replicate the logic of Algorithm 1.
> >
> > That being said, we agree with the reviewer that using a direct implementation of Algorithm 1 as a wrapper in many real situations would be impractical and easily detected, so we will change our language in Section 3.3 to reflect this and make the writing more clear.
> >
> > **Q2: “Does this imply that daze can persist into the next episode?”**
> >
> > Sorry if our answer caused some confusion here. We confirm that dazed states **do not** persist between episodes. Once a new episode starts the simulator draws a clean initial state $s \in S$ and resets the “daze” counter (in algorithm 1) to 0. This clean state can become a triggered state with probability $\beta$ as with any other state.
> >
> > In the case we discussed, the episode does indeed end once the agent *transitions* with respect to  “move-right” but not necessarily if they choose the action “move right” in a dazed or triggered state. To be precise, if the agent ignores the trigger and chooses “move right” in $\delta(I)$ they will transition with respect to “stay” with probability 0.5 (continuing the episode) or “move right” with probability 0.5 (ending the episode). This results in less return overall than choosing the target action "stay" in expectation.
> >
> > **C1: “Regarding $S$ and $S_\delta$ , as I understand it, treating them as either disjoint or overlapping can both be considered reasonable.”**
> >
> > Thanks for this feedback. We agree that both formulations are reasonable, though we believe there are additional ways that our formulation can also be stealthy in practice. For instance, in real world image based domains, the state/observation space may already contain all possible images, meaning $S$ and $S_\delta$ must overlap. In this case the trigger can be something rare, as you suggest, such as a particular QR-Code pattern or a strangely colored object that doesn’t appear naturally during training. This effectively maintains our theory while being more stealthy as the trigger function $\delta$ produces valid images without obvious perturbations. .
> >
> > **C2: Regarding the uniform distribution, I personally think it might be better not to restrict the distribution type.**
> >
> > Thanks for your input on this, but we are concerned that defining $\mathcal{U}$ as an arbitrary distribution may make our attack objectives less clear. We would like to clarify that Equation 1 merely represents a computation of the average Attack Success Rate across all states. Therefore, to maximize Equation 1 the adversary must maximize the effectiveness of their attack in all states, resulting in more versatile control at test time.

---

### Official Review · Reviewer_V5NT · 2025-10-27

**Soundness:** 3
**Presentation:** 3
**Contribution:** 2
**Rating:** 4
**Confidence:** 5

**Summary:**

This paper studies a new framework of attack called Dazed attack where the attacker essentially has control over state transition of the environment and can perturb the action of the agent using a malicious simulator. The authors propose this as a "reward-free" threat model which contrasts to prior works like TrojDRL, SleeperNets which rely on reward poisoning. The attack mechanism work by punishing agent on non-compliance(on not following the target action) by forcing them to a "Dazed" state where agents actual actions are ignored and an uniform policy is played on behalf of agent thereby leading to poor performance. The authors provide theoretical guarantees on their attack framework and demonstrate the effectiveness of it through empirical experiments on Atari and Mujoco environments.

**Strengths:**

1. The paper is well written and most of concepts are presented in an organized way.
2. The paper claims to present a novel threat model which I think is misleading - more on this in the weakness section.
2. The authors provide good theoretical guarantees on their attack algorithm.
3. The attack framework leads to high empirical success rate on both environments outperforming reward-poisoning counterparts.

**Weaknesses:**

1. The paper's central claim that this is "more constrained" or "weaker" form of attack is quite misleading. The argument is highly one-dimensional focusing only of lack of reward access but it ignores the fact that the attacker is granted the complete control over the environment transition and action taken by agent to random action. This is an extremely powerful attack capability and it is not convincingly argued why this is 'weaker' than simply altering a scalar reward.
2. The paper misses critical recent related works on state and action perturbations and the work should rightfully be compared to those papers rather than reward-poisoning attack papers. Since authors fail to compare against there more relevant baselines, the paper's positioning is not very clear and its claims on "weaker" form of attack appear highly overstated to the reviewer.
3. The state perturbation attacks are highly detectable and the authors have failed to discuss any defense mechanism in the paper.

References:

[1]. Optimal Attack and Defense for Reinforcement Learning, McMahan et. al, AAAI-24.

[2].  Robust Reinforcement Learning on State Observations with Learned Optimal Adversary, Zhang et. al, ICLR21. (Also look at their follow up works)

**Questions:**

1. Can you please provide a more robust justification of why your framework is "weaker"?
2. Why is related work, experiments section completely miss comparisons to prior works on state and action perturbation attacks?
3. For a defender, it would be easy to detect the presence of a dazed state by investing the fact that any action taken by agent there leads to similar poor performance, thereby declaring the environment to be malicious. How the attacker would protect themselves against such strong signal of identification?

---

> ### Author Response · Authors · 2025-11-15
> **Response to Reviewer V5NT**
>
> Hello Reviewer V5NT,
>
> Thank you for your insightful feedback and questions, we greatly appreciate it. In this comment we will respond to all of your questions and concerns. Feel free to ask more questions or ask for further clarification, we look forward to the discussion. After reading our rebuttal and the resulting discussion we kindly ask you to consider increasing your assessment of our paper, should you deem fit.
>
> **Concerns over the Weakness of the Daze Threat Model**
>
> Thank you for your feedback on this. We assert that Daze’s threat model is weaker than prior works because 1.) the attack requires less information (rewards) to be available to the adversary and 2.) the attack exploits realistic trust structures within the supply chain of DRL training.
>
> It is also important to remember that prior, reward poisoning methods make very strong assumptions for their attacks to be feasible. Specifically, direct action manipulation attacks, like TrojDRL, require access to **both** the agent’s actions/transitions along with their reward function. Therefore Daze, without altering or observing rewards, has a strictly weaker threat model.
>
> Additionally, attacks like SleeperNets and Q-Incept require malicious software with **RAM-level** access to be installed on the agent’s training machine to alter/learn from the agent’s training data. Therefore, we assert that reward manipulation alone requires multiple underlying and strong assumptions which are infeasible in many scenarios. By removing reward access we create a much weaker adversary operating under more challenging and realistic constraints. This proves that protecting one’s rewards is not enough to prevent backdoor attacks.
>
> **Comments Regarding Prior Action Manipulation Attacks**
>
> Thank you for highlighting these papers. The reason we do not compare against works like [1] and [2] is that they study test time attacks against fixed policies, not training time poisoning attacks against policies that are actively learning. They **do not** attempt to implant a backdoor at training time nor do they alter the agent’s training process. Therefore, the objectives of these papers are completely orthogonal to those of our work and the methods are incomparable.
>
> The baselines we compare against in our paper, TrojDRL, SleeperNets, and Q-Incept, represent the current state of the art for both action and reward manipulation backdoor attacks. We’ve added additional context to the revised draft to clarify this and will include references to [1] and [2] in the final draft.
>
> **“The state perturbation attacks are highly detectable”**
>
> We would like to push back on the assertion that state perturbation attacks are highly detectable. We first suggest the reviewer to read Appendix D.1 as it highlights the adversary’s ability to hide the Daze attack within a standard suite of domain randomization techniques [6]. This makes detection of Daze’s manipulations non-trivial, requiring new defense techniques to be studied.
>
> Furthermore, many works like [3], [4], and [5] have shown that backdoor triggers can be incredibly difficult to distinguish from real features in the data. Extending this reasoning to DRL: how is one to distinguish a green light, which always induces the “accelerate” action, from a white/red cross which does the same? Answering this is currently an open problem within the existing literature. Therefore, we hope attacks like Daze can further highlight the need to develop more robust trigger detection techniques in DRL.
>
> **“the authors have failed to discuss any defense mechanism…”**
>
> Our initial submission included discussion on defenses in Appendix D.3. In short, since Daze proposes a wholly new threat vector and attack methodology, there are no defenses currently designed specifically to prevent/mitigate the attack. While some works, like [7] and [8], do attempt to implement universal defense techniques, they can be easily broken with evasive triggers, as shown in prior work [9]. We agree that this discussion is important, therefore we have included further discussion on defenses in Section 5 of the revised text.
>
> **“For a defender, it would be easy to detect the presence of a dazed state…”**
>
> Similar to our discussion on detecting the trigger, we would like to push back against the assertion that Daze can be detected in this way. Again, as we discuss in Appendix D.1, it is fairly common for practical DRL implementations to use forms of domain randomization [6] to improve the agent’s robustness at test time. Specifically, many techniques will probabilistically force the agent to transition with respect to randomly sampled actions, irrespective of their chosen action. This benign technique is identical to the action manipulation of Daze making distinguishing between these two behaviors very difficult. We look forward to future works which may aim to solve this issue or others which make Daze even harder to detect.

---

> > ### Author Response · Authors · 2025-11-15
> > **Citations for the Above Response**
> >
> > [3] Khaddaj et al. "Rethinking backdoor attacks." ICML 2023.
> >
> > [4] Zhang et. al. Towards Backdoor Attacks against LiDAR Object Detection in Autonomous Driving. 533-547. 10.1145/3560905.3568539.
> >
> > [5] Tu et. al. “Physically realizable adversarial examples for lidar object detection” CVPR 2020
> >
> > [6] Tobin et. al. “Domain randomization for transferring deep neural networks from simulation to the real world” IROS 2017
> >
> > [7] Bharti et. al. “Provable defense against backdoor policies in reinforcement learning.” NeurIPS 2023
> >
> > [8] Chen et. al. “Bird: generalizable backdoor detection and removal for deep reinforcement learning” NeurIPS 2023
> >
> > [9] Rathbun et. al."Adversarial Inception Backdoor Attacks against Reinforcement Learning." ICML 2025

---

### Official Review · Reviewer_K4RV · 2025-11-01

**Soundness:** 3
**Presentation:** 3
**Contribution:** 3
**Rating:** 6
**Confidence:** 3

**Summary:**

The paper introduces a novel and highly practical threat model for backdoor attacks in Reinforcement Learning (RL). Instead of assuming an attacker can poison an agent's rewards (a common but strong assumption), this work considers a more subtle attacker: the developer of an "untrusted simulator". The paper provides formal proofs that this mechanism guarantees both attack success (the agent must learn the backdoor) and attack stealth (the agent's policy on the benign task remains optimal). Empirically, Daze is shown to outperform existing SOTA attacks (which require reward access) in continuous domains and match their performance in discrete domains.

**Strengths:**

Novel and Practical Threat Model: The paper correctly identifies that as RL becomes more reliant on third-party simulators (MuJoCo, PyBullet, etc.) and cloud services, the "untrusted simulator" is a major, realistic security blind spot. It is the first to formally define and attack the "reward-free" threat model, which is a significant contribution to the field of RL security.


Strong Theoretical Guarantees: The attack is not just a heuristic. The authors provide formal proofs (Theorem 1 and Theorem 2) that an agent optimizing its own returns must converge to a policy that is both successfully backdoored and stealthy. The detailed proofs in the appendix (Appendix E) appear sound.

**Weaknesses:**

1） Would the "Daze" attack successfully transfer in a complex, uncontrolled, "in-the-wild" environment that isn't almost identical to the training simulator?

2） The paper claims this assumption is "fairly weak". However, one could argue that in highly stochastic or complex environments, a random or exploratory action in a specific state might be part of a near-optimal policy. If the "punishment" of random actions isn't severe, the agent won't be as strongly incentivized to learn the backdoor, and the attack would fail.

**Questions:**

Please see the weakness

---

> ### Author Response · Authors · 2025-11-15
> **Response to Reviewer K4RV**
>
> Hello Reviewer K4RV,
>
> Thank you so much for your feedback and strong praise of our paper’s ”novel and practical threat model”, “strong theoretical guarantees” and our “significant contribution to the field of RL security”, we greatly appreciate it. In this comment we will respond to all of your questions and concerns. Feel free to ask more questions or ask for further clarification, we look forward to the discussion. After reading our rebuttal and the resulting discussion we kindly ask you to consider increasing your assessment of or confidence in our paper, should you deem fit.
>
> **“Would the "Daze" attack successfully transfer in a complex, uncontrolled, "in-the-wild" environment that isn't almost identical to the training simulator?”**
>
> Absolutely! First, as you mentioned, the Daze attack has strong theoretical guarantees when implanted into any generic MDP, which give us some certainty that the attack will remain effective even in complex environments.
>
> Furthermore, even as the task complexity scales up, the complexity of the targeted adversarial behavior, a single target action, remains fixed. This also remains true for cases where the “environment isn’t almost identical to the training simulator”. The environment dynamics may change, but the trigger and target action remain fixed. The challenge here is for the agent to maintain high return in benign states. Attack success, on the other hand, will be preserved in triggered states.
>
> **“The paper claims this assumption is "fairly weak". However, one could argue that in highly stochastic or complex environments, a random or exploratory action in a specific state might be part of a near-optimal policy.”**
>
> Thank you for this insightful comment. One thing we want to disentangle here is what an “exploratory action” is in this context. During the *training* phase we do agree that random exploration is a necessary component of RL methods more broadly, but it does not mean that uniformly random behavior is part of an optimal policy.
>
> Put in another way, our assumption requires that, under the optimal policy, there is a *strict* subset of *optimal* actions $A^* \subset A$ to choose from in each state. Therefore, when executing the optimal policy, there must always exists an action that performs worse than others. We believe this assumption is reasonable and applies to most tasks being researched today, especially continuous action space domains where maintaining a strictly equal value across all actions in $\mathbb{R}$ is highly unlikely. We have added further discussion on this topic in Section 3.3 of the revised submission.
>
> **"If the 'punishment' of random actions isn't severe, the agent won't be as strongly incentivized to learn the backdoor, and the attack would fail.”**
>
> We appreciate this comment, but believe there is more nuance here. Our reason for making Assumption 1 was to prove the optimality of the target action in *all* states $s \in S$. If randomly sampled actions are instead optimal for some subset of states $S_r \subset S$, then the target action will still be optimal in all other states $S$ \ $S_r$, as we have proven. Furthermore, since the agent remains “dazed” for multiple time steps, it is likely that the target action will still be optimal for states in $S_r$. As the agent continues to transition with respect to randomly sampled actions, it is likely they will enter some state in $S$ \ $S_r$ for which randomly sampled actions are sub-optimal. This makes any action other than $a^+$ strictly sub-optimal in the original, triggered state from $S_r$.
>
> Therefore, the attack will still remain effective even if our assumption is violated in a subset of states and will not fail outright. We believe that highlighting this discussion is important and have made it more clear in a new Section 3.3 of the updated submission.

---

### Official Review · Reviewer_7FdN · 2025-11-01

**Soundness:** 4
**Presentation:** 3
**Contribution:** 4
**Rating:** 10
**Confidence:** 3

**Summary:**

In this paper, the authors propose DAZE, a reward-free backdoor attack against RL agents. DAZE creates triggered and dazed states: when the agent encounters a trigger, it is forced to perform suboptimal actions. Agents that take the target action (i.e., the adversary's desired behavior) avoid a penalty during training; therefore, they learn to execute the backdoor behavior. Experimental results (in both continuous and discrete action spaces) show that DAZE achieves a high attack success rate while inducing no drop in benign return. The authors also validated DAZE on real robotic hardware.

I believe that the work is strong and well motivated. It explores simulator-level vulnerabilities in RL with a solid theoretical framework and experimental results.

**Strengths:**

S1. Novelty: The authors highlight realistic adversary capabilities when the simulator is malicious: no read or write access to the agent's rewards, or cannot alter the rewards. Therefore, their attack, DAZE, operates under constrained but realistic assumptions unlike the prior backdooring work with the reward manipulation assumption.

S2. Impactful Demonstration: In addition to empirical results in the MuJoCo and Atari domains, the real-world robotic example (also supplemented by a video) effectively shows the practical security implications of the attack.

**Weaknesses:**

W1. Practical Defenses: Although the attack is realistic, the authors have limited discussion on potential detection or mitigation strategies.

W2. Trigger Design: The visual or input-level triggers are somewhat artificial. These triggers might not go unnoticed during deployment.

**Questions:**

Q1. How sensitive is Daze to the shape of the trigger? Can a slightly perturbed trigger still activate the backdoor?

Q2. Page 8, line 419, type (ore -> or, Interesction -> Intersection)

**Details Of Ethics Concerns:**

No ethics concerns

---

> ### Author Response · Authors · 2025-11-15
> **Response to Reviewer 7fdN**
>
> Hello Reviewer 7FdN,
>
> Thank you for your review and high praise of our paper, we greatly appreciate it. Our goal in this response is to clarify any remaining questions or concerns you have with this work, hopefully increasing your confidence in our paper and its results. Please feel free to ask further clarifying questions should you have them. We look forward to discussing with you.
>
> **“Practical Defenses: Although the attack is realistic, the authors have limited discussion on potential detection or mitigation strategies.”**
>
> Thank you for your feedback on this. Since Daze proposes a new attack method and threat vector, there are currently no defenses which are directly applicable to defending against it. We do include some discussion on defenses in Appendix D.3, but we agree that further discussion is important. Therefore we added new Section 5 including a discussion of defenses in the revised submission.
>
> **“Trigger Design: The visual or input-level triggers are somewhat artificial. These triggers might not go unnoticed during deployment.”**
>
> This is a fair criticism of backdoor attacks more broadly, and particularly with vision-based agents. That being said, there is a lot of existing work on physically realizable and discrete triggers in the space of supervised learning [1] [2] which are likely applicable to DRL as well. One benefit of Daze in this regard is that it is agnostic to the design of the trigger function $\delta$. Therefore, even stealthy triggers like those proposed in prior studies will  work with Daze.
>
> **“How sensitive is Daze to the shape of the trigger? Can a slightly perturbed trigger still activate the backdoor?”**
>
> Based on our empirical results on real hardware, we found the trigger to be fairly robust. For instance, if you look at figure 6 in Appendix C.4 you can see that the real world trigger is often tilted, in a different position, or a different size than the trigger used during training. Despite the discrepancy, we were still able to reliably induce the target action on the real robots and cause the agents to crash. It is likely that future works can build further on the Daze framework to design attacks with even more robust or stealthy triggers.
>
> **“Page 8, line 419, type (ore -> or, Interesction -> Intersection)”**
>
> Thank you for pointing out these mistakes, we have fixed them in the revised draft of the paper.
>
> [1] Zhang et. al. Towards Backdoor Attacks against LiDAR Object Detection in Autonomous Driving. 533-547. 10.1145/3560905.3568539.
>
> [2] Tu et. al. “Physically realizable adversarial examples for lidar object detection” CVPR 2020

---

> > ### Comment · Reviewer_K4RV · 2025-11-25
> >
> > Thanks for your response! I will keep my score.

---

### Author Response · Authors · 2025-11-15
**Overview of Our Reviewer Response**

Hello Reviewers and Area Chairs,

Thank you for taking the time to review our paper and provide feedback. We strongly believe this advice has significantly improved the clarity and quality of our paper, making it easier to understand our novel method, threat model, and contributions. **All modifications we promise to each reviewer are currently present and highlighted with blue text in the revised submission, available above.** In short, we have made the following improvements to the paper:

* Figure 2 was added to visually represent and clarify our threat model in Section 2.2

* Section 2.4 “On the Necessity of Assumption 1” was included to add further details on the robustness of our method with respect to Assumption 1.

* Further context on our choice of baselines was added in Section 4, highlighting the action manipulation of TrojDRL and Q-Incept along with the dynamic reward poisoning of SleeperNets. This makes it clear that our baselines cover a wide range of fundamentally different attack approaches.

* Some sentences were added in Section 4.3 to highlight Daze’s improved performance over baselines in terms of BR score.

* Section 5 was added for further discussion on potential defense techniques against Daze.

* Grammar and spelling fixes were included along with other minor clarifications to further improve the quality of our presentation.

We also greatly appreciate the high praise our work received from the reviewers, with many highlighting the novelty and importance of our threat model, method, and theoretical results. The initial reviews have been very fruitful towards improving the quality of our paper, so we look forward to further feedback and discussion with the reviewers. Given the thoroughness of our responses and the many improvements we have made to the paper, we kindly ask the reviewers to consider reflecting this in their chosen scores or confidence values, should they deem fit.

Thank you,

Authors of Submission #9316

---

### Meta-Review · Area_Chair_44om · 2025-12-17

**Summary:**

This paper introduces DAZE, a novel backdoor attack against RL agents that operates under a constrained "reward-free" threat model where the adversary controls the simulator but cannot observe or modify rewards. Reviewers find the core threat model novel, realistic, and impactful, particularly for scenarios involving third-party simulators. The theoretical guarantees for attack convergence and stealth are seen as a significant strength, and the demonstration on a real-world robot is compelling. Though the rebuttal, the authors have addressed most of the comments from the reviewers, particularly on the clarification why DAZE is weaker than reward poisoning. Also and the necessity of Assumption 1. Also, they have revised the draft based on the reviewers' comments to make it more sound and solid. I think this work adds values to the area of adversarial RL, providing meaningful insights for how RL agents are negatively affected by backdoor attacks without altering or even observing their rewards.

**Reviewer Concerns:**

Reviewers 7FdN, K4RV, and R7TP concerns have been addressed by the rebuttal. However, the concerns from reviewer V5NT may not be fully addressed yet, but most of them have been resolved by the authors.

**Reviewer Scores:**

Based on the discussion, I think most of the reviewers would be willing to raise the score. Reviewer R7TP has declared that. While Reviewer K4RV has not discussed with the authors, given the responses, I think the score would be increased. Reviewer 7FdN remains the same score. For reviewer V5NT, the score might be lowered if authors did not adequately address the comparison to state/action perturbation literature.

---

### Decision · Program_Chairs · 2026-01-26

Accept (Poster)